# Differentially Private Image Classification by Learning Priors from Random Processes

**Xinyu Tang**[*]  **Ashwinee Panda**[*]  **Vikash Sehwag**  **Prateek Mittal**
Princeton University

## Abstract

In privacy-preserving machine learning, differentially private stochastic gradient descent (DP-SGD) performs worse than SGD due to per-sample gradient clipping and noise addition. A recent focus in private learning research is improving the performance of DP-SGD on private data by incorporating priors that are learned on real-world public data. In this work, we explore how we can improve the privacy-utility tradeoff of DP-SGD by learning priors from images generated by random processes and transferring these priors to private data. We propose DP-RandP, a three-phase approach. We attain new state-of-the-art accuracy when training from scratch on CIFAR10, CIFAR100, MedMNIST and ImageNet for a range of privacy budgets $\varepsilon \in [1, 8]$. In particular, we improve the previous best reported accuracy on CIFAR10 from $60.6\%$ to $72.3\%$ for $\varepsilon = 1$. Our code is available at https://github.com/inspire-group/DP-RandP.

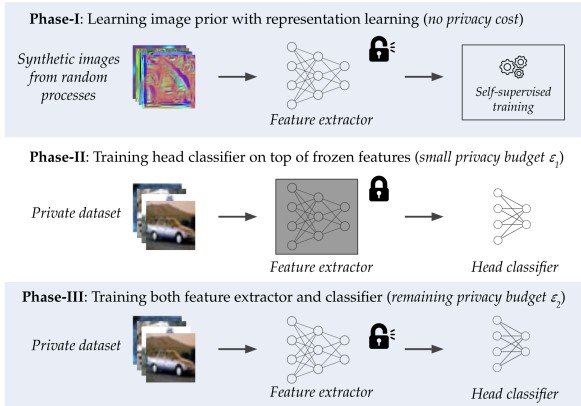

Figure 1: Our proposed DP training pipeline (DP-RandP) has three distinct phases. In Phase I, we sample images from random processes and train a feature extractor with representation learning to embed image priors beneficial for visual tasks. In Phase II, we spend a small privacy budget to train a linear classifier on top of extracted features of private data. In Phase III, we update all parameters with our remaining privacy budget. We demonstrate that incorporating image priors in Phase-I significantly improves DP training and adopting Phase II before training the whole network in Phase III can further improve test accuracy in DP training.

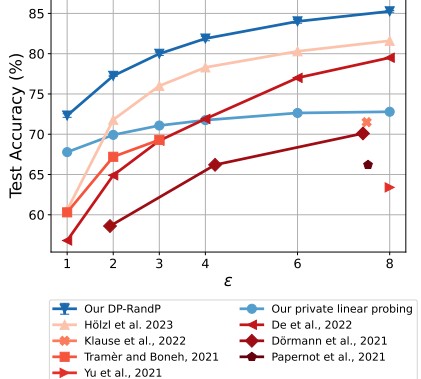

Figure 2: Comparing our results on CIFAR10 and previous state-of-the-art for $(\varepsilon, 10^{-5})$-DP setup. Our full DP-RandP outperforms all previous works on this commonly benchmarked dataset and reduces the privacy cost needed to achieve $80\%$ accuracy from $\varepsilon = 6$ to $\varepsilon = 3$. Even our private linear probing from noise prior outperforms all previous work at $\varepsilon = 1$ but crucially has diminishing returns, a shortcoming that we address with our proposed three-phase DP training framework (DP-RandP).

---

[*]Equal contribution

37th Conference on Neural Information Processing Systems (NeurIPS 2023).

# 1 Introduction

Machine learning models are susceptible to a range of attacks that exploit data leakage from trained models for objectives such as training data reconstruction and membership inference [58, 4]. Differential Privacy (DP) is the gold standard for quantifying privacy risks and providing provable guarantees against attacks [20, 21]. DP implies that the outputs of an algorithm e.g., the final weights trained by stochastic gradient descent (SGD) do not change much (given by the privacy budget $\varepsilon$) across two neighboring datasets $D$ and $D'$ that differ in a single entry.

**Definition 1 (Differential Privacy)** *A randomized mechanism $\mathcal{M}$ with domain $\mathcal{D}$ and range $\mathcal{R}$ preserves $(\varepsilon, \delta)$-differential privacy iff for any two neighboring datasets $D, D' \in \mathcal{D}$ and for any subset $S \subseteq \mathcal{R}$ we have $\Pr[\mathcal{M}(D) \in S] \leq e^{\varepsilon} \Pr[\mathcal{M}(D') \in S] + \delta$.*

Differentially Private Stochastic Gradient Descent (DP-SGD) [59, 1] is the standard privacy-preserving training algorithm for training neural networks on private data, with an update rule given by $w^{(t+1)} = w^{(t)} - \frac{\eta_t}{|B|} \left( \sum_{i \in B} \frac{1}{c} \texttt{clip}_c (\nabla \ell(x_i, w^{(t)})) + \sigma \xi \right)$ where the changes to SGD are the per-sample gradient clipping $\texttt{clip}_c(\nabla \ell(x_i, w^{(t)})) = \frac{c \times \nabla \ell(x_i, w^{(t)})}{\max(c, ||\nabla \ell(x_i, w^{(t)})||_2)}$ and addition of noise sampled from a $d$-dimensional Gaussian distribution $\xi \sim \mathcal{N}(0, 1)$ with standard deviation $\sigma$. DP-SGD introduces bias and variance into SGD and therefore degrades utility, creating a challenging privacy-utility tradeoff. For example, the state-of-the-art accuracy in private training is only $60.6\%$ on CIFAR-10 at $\varepsilon = 1$ [35], while Dosovitskiy et al. [19] obtains $99.5\%$ accuracy non-privately.

Theoretical analysis of the DP-SGD update yields that noise addition is especially harmful to convergence at the start of training [24], and that pretraining on public data can greatly improve convergence in this initial phase of optimization [46] by providing a better initialization [25]. Previous works have improved the privacy-utility tradeoff of DP-SGD by pre-training on large publicly available datasets, such as ImageNet [16], to learn visual priors [50, 49, 54, 9]. Other works assume that a small subset of in-distribution data is publicly available [68, 2, 45, 52].

Interestingly, previous work has uncovered that synthetic data learned from random processes [44, 10, 22, 38] can be used in representation learning [30, 12, 11] to learn highly useful visual priors [39, 5]. Because there is a large distribution shift between synthetic images and natural images, training on synthetic images does not incur any privacy cost. In this work, we leverage noise priors learned from synthetic images to boost the performance of DP training, and make the following key contributions:

- We find that while noise priors are only marginally helpful in non-private training, we can unlock their full potential to improve private training with a carefully calibrated training framework.

- We provide empirical evidence that priors learned from random processes become more critical as the privacy budget decreases, because priors provide fast convergence at the start of training, and have a much larger impact on private training than non-private training.

- We find that training a single linear layer (linear probing) on top of a pretrained feature extractor that has learned noise priors is more robust to large amounts of noise addition than end-to-end training of the entire network. We demonstrate this insight by linear probing with a small privacy budget $\varepsilon = 0.1$ to $57.1\%$ on CIFAR10, achieving nontrivial performance with lower privacy cost than previous work has considered.

- We observe that while linear probing from noise prior has diminishing returns on performance as the privacy budget increases, end-to-end training of the entire network continues to improve with large $\varepsilon$ but critically struggles for small privacy budgets.

- We harness our insights by proposing a privacy allocation strategy that combines the benefits of learning from priors, linear probing, and full training into our full method DP-RandP (Visualized in Fig. 1). Our proposed approach pretrains a feature extractor on synthetic data to learn priors from random processes without paying privacy cost, then pays a small privacy cost for linear probing and makes the best use of our remaining privacy budget by updating all parameters to adapt our learned features to the private data.

- We evaluate DP-RandP against previous work and unlock new SOTA performance across CIFAR10, CIFAR100, MedMNIST and ImageNet across all evaluated privacy budgets $\varepsilon \in [1, 8]$. We provide a snapshot of our results in Fig. 2.

## 2 DP-RandP: Mitigating the effects of noise in DP-SGD with noise prior

**Synthetic image generation without natural images.** Recent progress in computer vision shows that pretraining models on synthetic images without natural images [5, 6, 39] can learn visual priors that are competitive to priors from natural images. The synthetic images can either be generated from texture and fractal-like noise [6, 39], or structure priors extracted from an untrained StyleGAN [5].

**Noise prior from synthetic images for private training.** We consider pretraining on synthetic data, generated by random processes [5, 6] (example images are in Fig. 3), as a *noise prior* in differentially private training. We use contrastive representation learning [12, 11, 30, 64], which aims to learn features invariant to common image transformations, to learn good visual features using synthetic images. At a high level, rather than using a random or 'cold' initialization for our downstream private dataset, we are using representation learning to obtain a 'warm initialization' that encodes priors learned from random processes. Across common natural vision tasks, noise priors also ensure that there is no privacy leakage from the pre-training data into this 'warm initialization'.

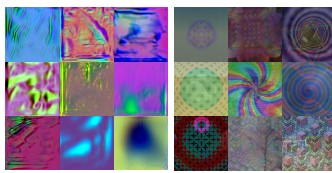

(a) StyleGAN [5]   (b) Shaders [6]

Figure 3: We use synthetic data generated from random processes in representation learning to learn useful image priors. Due to the high distribution shift from real images, we do not incur privacy costs when learning noise priors from these images.

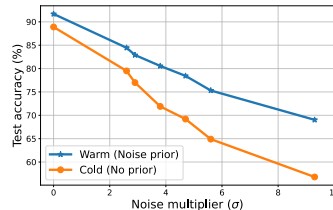

Figure 4: Comparison of training from a random initialization (cold) and pre-trained encoder on synthetic dataset (warm) across different privacy budgets (the x-axis is the corresponding $\sigma$).

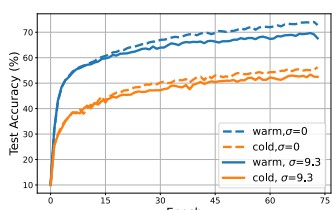

Figure 5: We zoom in on $\sigma = 0, \sigma = 9.3$ with learning rate$= 0.4$ and find that the improvement of the warm start over cold start is mostly due to the initialization using synthetic dataset.

**How much noise prior benefits private training?** As an initial exploration, we obtain a warm initialization via representation learning on synthetic images and compare it with a competitive DP-SGD baseline [15] that uses cold (random) initialization (Fig. 4). We find that while the warm initialization only improves performance by $2.7\%$ when $\sigma = 0$, i.e., non-private training, the warm initialization improves performance by $12.5\%$ when $\sigma = 9.3$ (equivalent to $\varepsilon = 1$). In Fig. 5, we zoom in on a single point of comparison between $\sigma = 0$ (dashed) and $\sigma = 9.3$ (solid) for warm and cold initializations. We find that over the course of training, as more noise is added, both the $\sigma = 9.3$ warm and cold initializations not only converge at the same rate (given an appropriately small learning rate) but also diverge from the non-private runs at similar rates. *Although warm and cold start obtain different results in the private setting, the difference is mostly due to the initialization and is therefore magnified at smaller privacy budgets.*

We support this claim by drawing a connection to previous works. Ganesh et al. [25] prove the existence of out-of-distribution public datasets that can achieve small test loss on target private datasets. Bu et al. [8] prove that in the gradient flow setting for the NTK regime, that is, when we are taking very small steps $\eta \to 0$, noise does not impact convergence. Mehta et al. [50], Panda et al. [54] propose scaling the learning rate $\eta$ inversely with the noise $\sigma$. We combine our analysis with these previous works by noting that if we start training from fixed initialization and set the learning rate near-zero for small privacy budgets [50, 54], we enter the regime [8] where the noise does not impact convergence, and achieving nontrivial performance for these small privacy budgets provides the first empirical evidence for the theory [25] that *only the initialization matters*.

We gather our insights into a design goal that will enable us to achieve nontrivial performance under strict privacy constraints. We want to encode a learned prior into our initialization and then adapt this prior to private data. We now introduce DP-RandP, a method that achieves this key design goal.

## 2.1 Three-phase differentially private training framework

We propose a three-phase DP training framework DP-RandP that has two phases after pretraining on synthetic images, and significantly improves the performance of DP training (Fig. 1). We first learn noise priors by training a feature extractor on synthetic data (Phase-I). We then split our private training into 1) Learning the head classifier with frozen features (Phase-II) and 2) End-to-end training of the entire network to co-adapt the feature extractor and head classifier (Phase-III).

Our design is motivated by the strengths and weaknesses of linear probing and end-to-end training in the private setting. First, the $\ell_2$ norm of the Gaussian noise added to gradients in each DP-SGD step scales with the number of parameters in the model. Because the head classifier typically has far fewer parameters than the feature extractor, updating this linear layer reduces the amount of added noise. Second, there is a large distribution shift between synthetic data from random processes and natural images found in private datasets. Linear probing merely inherits the frozen pre-trained features, but end-to-end training of all layers can improve the pre-trained features by adapting them to the private dataset. Because adapting to our private dataset may require many end-to-end training steps, we improve the convergence by first learning a task-specific linear classifier that can be trained quickly, and then training the entire network end-to-end. DP-RandP satisfies our previously stated design goal by first obtaining a good initialization with pretraining, then transferring the prior encoded in this initialization to the private dataset with fast-converging linear probing, and finally end-to-end training to adapt to the private dataset as much as possible for a given privacy budget.

We further consider the design choice of how to allocate the privacy budget for the two private training phases ($\varepsilon_1$ and $\varepsilon_2$). Recall that our model is pre-trained on synthetic data in Phase I, thus it doesn't incur any privacy cost. In Phase II, we use DP-SGD with privacy budget $\varepsilon_1$ to train the linear classifier. We observe diminishing returns in accuracy with increasing $\varepsilon_1$, so we spend a smaller privacy budget on linear classifier training and allocate the rest to training the full network in Phase III. Allocating a higher budget to full training does not have diminishing returns; consider that $\varepsilon_2 = \infty$ we recover non-private accuracy. We now rigorously evaluate the performance of DP-RandP.

# 3 Evaluation

To outline the evaluation section we first overview our experimental setup and then evaluate the performance of DP-RandP on CIFAR10/CIFAR100/DermaMNIST in Sec. 3.2. We find that our method outperforms all previous works across multiple datasets, architectures, and privacy budgets. In Sec. 3.3 we find that our principled allocation of privacy budget $\varepsilon_1, \varepsilon_2$ between linear probing and end-to-end training is robust to different choices of $\varepsilon_1$ and $\varepsilon_2$. We also find that the private linear probing is a strong computationally efficient baseline. Specifically, we provide a new SOTA on ImageNet with private linear probing. We next provide a quantitive comparison of DP-RandP to DP with public data. Finally we discuss the computational costs of our method and find that DP-RandP can provide computational savings over previous methods.

## 3.1 Experimental setup

**Learning priors from images generated by random processes with representation learning.** In Phase I we sample images from StyleGAN-oriented [5], and train a feature extractor on these synthetic datasets with representation learning [12, 64] with the loss function proposed in Wang and Isola [64]. Although our method can accommodate any kind of synthetic data and representation learning method, we focus our evaluation on these datasets and methods. We consider other kinds of synthetic data and other representation learning in Appendix A.

**Datasets and models.** We evaluate DP-RandP on CIFAR10/CIFAR100 [41], DermaMNIST in MedMNIST [65, 66] and private linear probing version of DP-RandP on ImageNet [16]. For CIFAR10/CIFAR100, we use WRN-16-4 following De et al. [15]. For MedMNIST, we use ResNet-9 following Hölzl et al. [34]. We choose WRN-16-4 and ResNet-9 because these architectures for the corresponding datasets achieve the most compelling results in previous works [15, 34]. For ImageNet, we use a ViT-base [19] feature extractor pretrained on Shaders-21k [6] provided by Yu et al. [71] and train a linear classifier. We provide the results on CIFAR10, CIFAR100, DermaMNIST in Sec. 3.2 and results on ImageNet in Sec. 3.3. We also report results of WRN-40-4 on CIFAR10 in Appendix B.

**Implementation details.**   To ensure a fair comparison with previous works [15, 57, 35], we use standard DP-SGD [1] and make use of multiple data augmentations and exponential moving average (EMA) as proposed by De et al. [15], that are now standard techniques used in DP-SGD work [57, 35]. We allocate small privacy budget $\varepsilon_1$ to Phase II and remaining privacy budget $\varepsilon_2$ to Phase III according to the strategy detailed in Sec. 3.3. We report our results across different privacy costs and use $\delta = 10^{-5}$ for CIFAR10/CIFAR100/DermaMNIST by following previous works [15, 34].[2] When we report results, we report the standard deviation and accuracy averaged across 5 independent runs with different random seeds. We report implementation details for CIFAR10/CIFAR100/DermaMNIST in Appendix C and for ImageNet in Appendix G.

## 3.2   Evaluation of DP-RandP

We report the results of DP-RandP in Tab. 1, Tab. 2 and Tab. 3 for CIFAR10, CIFAR100 and DermaMNIST datasets, respectively.

**DP-RandP outperforms all previous works across all privacy budgets.**   In Tab. 1 we find that DP-RandP obtains higher performance than previous works [15, 35, 61, 40, 18, 69, 55] on CIFAR10 across the standard evaluated privacy budgets $\varepsilon \in [1, 8]$.

We first compare DP-RandP to De et al. [15] as our CIFAR10 model, optimizer and hyperparameters follow De et al. [15]; the only difference is the use of Phase I and Phase II in DP-RandP to learn a prior from synthetic data and allocate a small privacy budget to linear probing. Crucially DP-RandP outperforms De et al. [15] by *more than* 15% for the important privacy budget $\varepsilon = 1$.

Tramèr and Boneh [61] use a ScatterNet [53] to encode invariant image priors and Hölzl et al. [35] use equivariant CNNs [14] to learn transform invariant features. DP-RandP shares the same intuition of leveraging invariant image priors as these works [61, 35]. Instead of leveraging model architecture design for invariant features, we achieve this intuition by learning priors from images generated from random processes and design our three-phase framework to optimize use of this prior. Although DP-RandP shares this intuition of leveraging invariant image priors, DP-RandP achieves 12% improvement over previous works [61, 35] who both achieve 60% at $\varepsilon = 1$. Our improvement comes both from leveraging the priors from synthetic images and our design of three learning phases that makes the best use of priors. We provide a detailed comparison to Tramèr and Boneh [61] in Sec. 4 where we explain why and how the feature prior we learn from synthetic data provides better results than the feature prior provided by their architectures.

Table 1: Test accuracy (%) of DP-RandP and comparison to previous work on CIFAR10. Not shown in this table are Klause et al. [40], Dörmann et al. [18], Papernot et al. [55], Yu et al. [69] because they achieve 71.5% at $\varepsilon = 7.5$ ,70.1% at $\varepsilon = 7.42$, 66.2% at $\varepsilon = 7.53$ and 63.4% at $\varepsilon = 8$ respectively, that are not on the pareto frontier of previous work.

| Method | $\varepsilon = 1$ | $\varepsilon = 2$ | $\varepsilon = 3$ | $\varepsilon = 4$ | $\varepsilon = 6$ | $\varepsilon = 8$ | $\varepsilon = \infty$ |
|---|---|---|---|---|---|---|---|
| Tramèr and Boneh [61] | 60.3 | 67.2 | 69.3 | – | – | – | 73.8* |
| De et al. [15] | 56.8 | 64.9 | 69.2 | 71.9 | 77.0 | 79.5 | 88.9 |
| Hölzl et al. [35] | 60.59 | 71.86 | 75.96 | 78.27 | 80.26 | 81.62 | – |
| DP-RandP | $72.32_{0.22}$ | $77.25_{0.07}$ | $79.99_{0.21}$ | $81.88_{0.27}$ | $84.01_{0.23}$ | $85.26_{0.11}$ | 91.69 |

**CIFAR100 results.**   Previous work has not provided results on training from scratch for CIFAR100, perhaps because as we find in Tab. 2 the DP-SGD baseline following De et al. [15] does not perform well for $\varepsilon = 3$. We believe this to be a competitive baseline, but DP-RandP outperforms it by more than 10% across $\varepsilon \in [3, 8]$. Although CIFAR10 and CIFAR100 are both benchmark computer vision datasets, we find CIFAR100 to be a much more challenging task for private learning. One possible

---

[2]We run experiments for $\varepsilon = \infty$ with per-sample gradient clipping but without noise, because we are interested in the ability of DP-RandP to mitigate variance. We note that there is accuracy degradation from the non-private baseline even at $\varepsilon = \infty$ due to the bias introduced by per-sample gradient clipping [37]. When we report $\varepsilon = \infty$ results from previous work we mark them with $*$ when previous work does not report whether the non-private baseline uses clipping.

explanation for this is that private classifiers struggle to distinguish between many classes because the added noise is more likely to shift the decision boundary. We use the same hyperparameters for CIFAR10 and CIFAR100, that are likely suboptimal for CIFAR100, and the performance gap between $\varepsilon = 8$ and $\varepsilon = \infty$ may be mitigated if we train on CIFAR100 for longer than we do on CIFAR10. We encourage the use of CIFAR100 as a standard benchmark for private learning in the future because we find limited room for improvement on CIFAR10. In particular, the private and non-private gap between $\varepsilon = 8$ and $\varepsilon = \infty$ for CIFAR10 is only $\approx 6\%$ for DP-RandP.

Table 2: Test accuracy (%) of DP-RandP and comparison to DP-SGD baseline on CIFAR100.

| Method | $\varepsilon = 3$ | $\varepsilon = 4$ | $\varepsilon = 6$ | $\varepsilon = 8$ | $\varepsilon = \infty$ |
|---|---|---|---|---|---|
| DP-SGD | 30.73 | 34.45 | 39.66 | 44.22 | 66.68 |
| DP-RandP | $43.33_{0.15}$ | $46.40_{0.31}$ | $51.53_{0.13}$ | $55.02_{0.21}$ | 71.68 |

**DermaMNIST results.** We have shown that DP-RandP outperforms previous work on the standard CV benchmarks of CIFAR10 and CIFAR100, and now consider the privacy sensitive medical dataset DermaMNIST. Although CIFAR is a standard CV benchmark, there is limited previous work that evaluates on privacy sensitive data in CV such as medical images. In Tab. 3 we find that DP-RandP achieve improvements from $\varepsilon = 1$ to $\varepsilon = 7.42$ by up to $2.78\%$ over the DP-SGD baseline that we evaluate. Also, DP-RandP can achieve similar accuracy at $\varepsilon = 4$ as the result of DP-SGD at $\varepsilon = 7.42$, which reduces the privacy cost from $\varepsilon = 7.42$ to $\varepsilon = 4$. We also include the results of Hölzl et al. [34], that use equivariant neural networks [14]. DP-RandP is a uniform framework applicable to any model architectures. We leave the systematic investigation of more model architectures, such as equivariant neural network [14, 35] in DP-RandP, for future work.

Table 3: We follow previous work [34] and report the validation accuracy (%) of DP-RandP on DermaMNIST. We also report the test accuracy in Appendix D.

| $\varepsilon$ | $\varepsilon = 1$ | $\varepsilon = 4$ | $\varepsilon = 7.42$ | $\varepsilon = \infty$ |
|---|---|---|---|---|
| baseline in Hölzl et al. [34] | − | − | 72.41 | 78.48* |
| best in Hölzl et al. [34] | − | − | 74.17 | 77.84* |
| DP-SGD we evaluated | $69.00_{0.37}$ | $71.78_{0.87}$ | $74.08_{0.41}$ | 77.27 |
| DP-RandP | $71.78_{0.40}$ | $74.82_{0.55}$ | $75.91_{0.29}$ | 79.26 |

We find that the prior we learn from synthetic data is applicable to standard benchmarks and medical images. One direction for future work is to validate the robustness of this prior across more datasets.

## 3.3 Allocating privacy budget in DP-RandP

In this subsection we analyze the privacy budgets of Phase II ($\varepsilon_1$, linear probing) and Phase III ($\varepsilon_2$, full training) and how to allocate the overall privacy budget $\varepsilon$ among Phase II and Phase III. Specifically, we will present a new STOA on ImageNet with additional designs on linear probing.

In Fig. 6 we observe that DP-RandP is robust to the key algorithmic design choice of how much privacy budget to allocate to Phase II and Phase III (Please check Apendix E for more details of Fig. 6 and more results on different $\varepsilon$).

In particular, we find that even the worst choice of $\varepsilon_1/\varepsilon$ provides results comparable to the previous SOTA on CIFAR10 at $\varepsilon = 1$.[3] We first investigate the behavior at each extrema and conclude a general allocation strategy that provides good performance across CIFAR10, CIFAR100 and DermaMNIST.

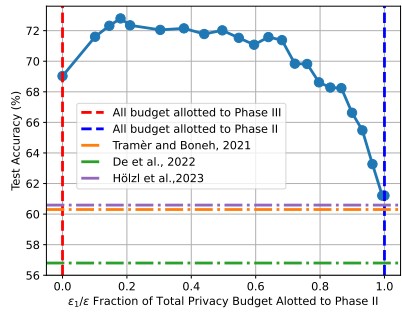

Figure 6: The fraction of total privacy budget allotted to Phase II for $\varepsilon = 1$. The performance is stable across a wide range of value [0.1, 0.4]. However, skipping either Phase-II or Phase-III leads to suboptimal test accuracy in DP training.

---

[3]Due to computational constraints we did not calculate error bars for Fig. 6, resulting in some nonconvexity.

**Allocating the entire privacy budget to Phase II is competitive for small privacy budgets but provides diminishing returns.** This corresponds to the right extreme in Fig 6 ($\varepsilon_1/\varepsilon = 1$) and is equivalent to doing linear probing on top of extracted features. We follow the training recipe in Panda et al. [54] and report the result of private linear probing in Tab. 4 for CIFAR10 and Tab. 5 for ImageNet [16] respectively ($\delta = 7.8 \times 10^{-7}$ is by $\delta = 1/|D_{train}|$ for ImageNet). We provide the detailed experimental set-up for CIFAR10 in Appendix F and ImageNet in Appendix G.

**CIFAR10 results.** We present the result of private linear probing on CIFAR10 by including more conservative privacy constraints like $\varepsilon = 0.03$. Our private linear probing can achieve $57.10\%$ at $\varepsilon = 0.1$, that is comparable to the result of $\varepsilon = 1$ in De et al. [15] that fully trains a WRN-16-4. Notably, previous work [61] also trains a neural network on top of the handcrafted features using an untrained ScatterNet [53]. DP-RandP is slightly better than their result at $\varepsilon = 3$. We can see that our non-private baseline ($74.05\%$) is slightly better than the non-private baseline ($73.8\%$) in Tramèr and Boneh [61], that shows that our feature extractor is better than the ScatterNet and therefore improves the performance under DP-SGD. We include a detailed comparison to Tramèr and Boneh [61] in Sec. 4. However, this private linear probing variant of DP-RandP can only achieve $74.05\%$ with no noise added, that is lower than the result at $\varepsilon = 4$ in De et al. [15], that shows that allocating all privacy cost to Phase II is a sub-optimal design choice.

Table 4: Test accuracy (%) of our private linear probing and comparison to previous SOTA for private learning on top of extracted features [61] on CIFAR10. Note that, our result is training a linear layer on top of the extracted features. The result of previous SOTA is by training a CNN on top of extracted features. Tramèr and Boneh [61] also report the private linear probing result $67.0\%$ at $\varepsilon = 3$.

| $\varepsilon$ | 0.03 | 0.1 | 0.2 | 0.5 | 1 | 2 | 3 |
|---|---|---|---|---|---|---|---|
| SOTA | - | - | - | - | 60.3 | 67.2 | 69.3 |
| Ours | 40.64 | 57.10 | 60.89 | 65.10 | 67.78 | 69.92 | 71.08 |
| (Std.) | 2.59 | 0.42 | 0.26 | 0.21 | 0.17 | 0.09 | 0.14 |

Table 5: Test accuracy (%) of our private linear probing with additional designs on ImageNet.

| $\varepsilon$ | 1 | 8 |
|---|---|---|
| De et al. [15] | - | 32.4 |
| Sander et al. [57] | - | 39.2 |
| Ours (ViT) | 26.54 | 39.39 |
| (Std.) | 0.11 | 0.03 |

**ImageNet results.** We achieve a new SOTA results on ImageNet. We achieve $39.39\%$ accuracy at $\varepsilon = 8$ and the previous SOTA [57] is $39.2\%$ at $\varepsilon = 8$. We use a Vit-base [19] model pretrained on Shaders-21k (the model checkpoint is provided by Yu et al. [71]). If we directly do Phase II with extracted features, we can achieve around $33\%$ accuracy at $\varepsilon = 8$, that is comparable to De et al. [15]. The direct linear probing result shows that linear probing (Phase II only after pretrained on synthetic data) is not enough for difficult tasks like ImageNet, and therefore updating full parameters in Phase III is necessary. While training the full ViT model on large datasets like ImageNet needs much computation resources (for example, Sander et al. [57] used 32 A100 GPUs), we could leverage the core concept of DP-RandP and adapt it to a computationally efficient version, i.e., we train a linear layer with additional modifications. We then obtain a new SOTA result of $39.39\%$ on the ImageNet-1k validation dataset. We now briefly explain the two main modifications of DP-RandP to achieve $39.39\%$. We provide a detailed explanation in Appendix G. *We emphasize that these modifications are more for computational efficiency; if we had enough compute to do full fine-tuning of the ViT on ImageNet with sufficiently large batch size, our original DP-RandP would still work.*

Our first modification is to approximate full fine-tuning by linear probing on larger feature representations that we create by aggregating intermediate representations from the network. This is because each block of vision transformers learns a different representation. However, linear probing only takes the representation from the penultimate layer as input and therefore the final representation may not be sufficient to learn the task. The representation of the input image after each block in the ViT has both a temporal and feature dimension, so we pool over the temporal dimension to gather a feature map of size (4, feature size). We concatenate the feature map of different blocks into a one-dimensional vector. By doing linear probing on these much larger features, we can also take advantage of intermediate representations.

Our second modification is to approximate the work of a LayerNorm or other normalization layer that we would update during full fine-tuning of the entire ViT, by manually normalizing the features.

To do this we first normalize each feature vector to a fixed norm. We next privately estimate the mean over the entire ImageNet feature vector dataset, using the Gaussian mechanism with a small privacy cost, and subtract the private mean from all feature vectors. This is equivalent to doing non-private centering and then adding the same Gaussian noise to the entire dataset. This procedure can be thought of as a one-time approximation to the normalization layer, which is known to speed up training by centering the data.

Within the two modifications of direct linear probing, we improve upon previous SOTA [57] and achieve 39.39% at $\varepsilon = 8$. This method is computationally efficient and one run can be done on a single A100 GPU in a few hours. We also provide the result for a stronger privacy guarantee, i.e., 26.54% at $\varepsilon = 1$.

**Allocating the entire privacy budget to Phase III struggles for small privacy budgets.** We report DP-RandP without Phase II on CIFAR10 in Tab. 6 (equals to $\varepsilon_1/\varepsilon = 0$ in Fig. 6). The result in Tab. 6 is slightly worse than DP-RandP in Tab. 1 and the utility gap between Tab. 6 and Tab. 1 decreases as $\varepsilon$ increases, that justifies the importance of Phase II in DP-RandP. Moreover, the result of DP-RandP without Phase II is significantly better than previous SOTA [15, 35, 61], that shows that the feature extractor pretrained on images from random process can capture the image prior.

Table 6: Fully privately training a WRN-16-4 with warm initialization. Test accuracy on CIFAR10.

| $\varepsilon$ | $\varepsilon = 1$ | $\varepsilon = 2$ | $\varepsilon = 3$ | $\varepsilon = 4$ | $\varepsilon = 6$ | $\varepsilon = 8$ | $\varepsilon = \infty$ |
|---|---|---|---|---|---|---|---|
| Accuracy(%) | $69.03_{0.23}$ | $75.31_{0.28}$ | $78.44_{0.19}$ | $80.56_{0.12}$ | $82.90_{0.10}$ | $84.45_{0.09}$ | $91.69$ |

**A general privacy budget allocation strategy.** We have observed that allocating the entire privacy budget to linear probing or full training is suboptimal. We now propose a simple yet effective general strategy to allocate the privacy budget. For small $\varepsilon$ ($\varepsilon \ll 1$), we set $\varepsilon_1 = \varepsilon$ to allocate the entire privacy budget to Phase II. This is because allocating the entire privacy budget to linear probing is competitive for small privacy budgets. As $\varepsilon$ increases, we decrease $\varepsilon_1/\varepsilon$. This is because as $\varepsilon$ increases, the noise multiplier will decrease. Therefore, it is easier to train the linear probing layer as $\varepsilon$ increases, and the percentage of total steps allocated to Phase II can be reduced to train a good linear probing layer and we can use the remaining steps for Phase III. Because the closed-form computation of $\varepsilon$ with numerical privacy loss distribution accounting by Gopi et al. [27] is challenging, we implement this strategy by using a fixed number of steps $n$ for linear probing in CIFAR10, CIFAR100, DermaMNIST, and increasing the number of steps in Phase III as $\varepsilon$ increases. We present the privacy allocation $\varepsilon_1/\varepsilon$ on CIFAR10 in Appendix E for results in Tab. 1, which validates this strategy.

### 3.4   In comparison to DP with public data

A major direction in improving the privacy utility trade-off for DP-SGD is by incorporating priors that are learned on real-world public data [68, 2, 45, 52, 50, 49, 54, 56]. These priors are either from small in-distribution data [68, 2, 45, 52] or large scale public data [50, 49, 54, 56].

Table 7: Comparison of DP-RandP with methods using public data. Test accuracy (%) on CIFAR10.

| Method | Model | Public Data | $\varepsilon = 1$ | $\varepsilon = 2$ | $\varepsilon = 4$ | $\varepsilon = 6$ | $\varepsilon = 8$ |
|---|---|---|---|---|---|---|---|
| Nasr et al. [52] | WRN-16-4 | 4% ID | 72.10 | 75.10 | 77.9 | 80.0 | – |
| Mehta et al. [50] | ViT-B/16 | ImageNet | 95.10 | 95.10 | 95.10 | – | 95.20 |
| Mehta et al. [49] | ViT-G/16 | JFT | 98.80 | 98.80 | 98.83 | – | 98.84 |
| Panda et al. [54] | beitv2 | ImageNet | 99.00 | – | – | – | – |
| DP-RandP | WRN-16-4 | Synthetic | 72.32 | 77.25 | 81.88 | 84.01 | 85.26 |

We present a quantitative comparison of DP-RandP and DP with public data works [54, 50, 49, 52] in Tab. 7. Our DP-RandP is comparable with Nasr et al. [52], which in fact utilizes a limited amount of in-distribution data as public data for pre-training. This indicates that the prior learned from images

generated from random processes can help as much as the prior learned from limited in-distribution public data. Compared to works [50, 49, 54] with access to large public data, there is still a gap between our DP-RandP and these works. Note that we consider a different threat model to this line of work where we do not have access to public data [62] and must instead make the best possible use of images drawn from random processes. Closing the gap between leveraging synthetic data and leveraging large-scale real public data is an interesting direction for future work.

### 3.5 Computational cost

DP-RandP consists of three phases. Some phases can be done in a single-run per dataset. For example, for feature extractor in Phase I, we only need to train a single feature extractor once for each evaluated dataset as the synthetic dataset and model architectures are the same. Also, for the private linear probing (LP). We report the computation cost in Tab. 8 for single-run per dataset. For Phase II and Phase III, we need to go through these processes at each run of our training process. Also, after we get the extracted features for LP, we need to train a linear layer each time we run the experiments. We report the computation cost in Tab. 9 for these procedures.

De et al. [15] also needs to fully train a WRN-16-4 and the major additional computation cost for DP-RandP is Phase I compared to De et al. [15]. However, the training in Phase I is done once for CIFAR10. Moreover, we can use the same feature extractor from Phase I for CIFAR10 and CIFAR100.

Table 8: One-time per dataset computational cost on CIFAR10. These procedures only need to be done once for each evaluated dataset. Phase I can be shared for CIFAR10 /CIFAR100.

|  | Phase I | feature extraction in LP |
| --- | --- | --- |
| Time | 16 h | 1 min |

Table 9: Computational cost on CIFAR10 for hyper-parameters given in Appendix C. These procedures are comparable to the standard training procedure such as De et al. [15].

|  | linear probing in LP | Phase II | Phase III |
| --- | --- | --- | --- |
| Time | 1 min | 12min | 5.5 h |

For the private linear probing experiment, each run of training a linear layer can be finished in 1 minute for CIFAR10 and 320 minutes for ImageNet while fully privately training a model costs much more time. A single run to privately train a WRN-16-4 for CIFAR10 takes around 5.5 hours for 875 steps with 1 A100 GPU in our evaluation. Also, as reported in previous work [57], a single run for ImageNet experiments needs to take four days using 32 A100 GPUs.

## 4 Discussion and related work

In this section we first provide a detailed comparison to previous work [61] that also uses priors to improve DP-SGD image classification. We then give an overview of the broader body of work on improving the privacy utility tradeoffs in DP-SGD. Finally we discuss the previous work [43] that also uses the two-stage training with domain-specific data for a different reasoning.

**Discussion on DP-RandP and Tramèr and Boneh [61].** Tramèr and Boneh [61] find that training a neural network (linear layer or CNN) on top of 'handcrafted' ScatterNet [53] features outperforms private 'deep' learning. While this method performs well for smaller values of $\varepsilon$, the non-private accuracy is limited because the features cannot be adapted to the private data. There are two key differences between DP-RandP and Tramèr and Boneh [61]. In Phase I we use representation learning to train the feature extractor on images sampled from random processes, to learn the prior that extract transformations-invariant features. Our feature extraction process is therefore much more general, and while we explore the use of different synthetic data, model architectures, and representation learning methods, there are many more methods in each of these categories that we have not explored. The second difference is between the training on top of extracted features, that is used by Tramèr and Boneh [61] and the private linear probing that we consider as a baseline in Fig. 2, and the combination of linear probing and full training that we use in DP-RandP. Comparing the improvements between DP-RandP and Tramèr and Boneh [61] confirms that our empirical improvements are mostly due to the advantage of DP-RandP over DP linear probing. In particular, for $\varepsilon = 3$, exchanging the handcrafted features of Tramèr and Boneh [61] for the pretrained feature extractor we use only improves performance by a modest $\sim 2\%$. However, our full DP-RandP exhibits more than $10\%$

improvement. Our innovation over Tramèr and Boneh [61] is therefore twofold: we introduce the potential of pretraining on synthetic data for the DP community, and also provide guidance on how to better transfer features learned from synthetic data to private training.

**DP with public data.** A major direction in improving the privacy utility tradeoff in DP-SGD for image classification is the principled use of public data. Several works [68, 2, 45, 52] make use of public data under a different threat model by treating a small fraction of the private training dataset as public. There is also another line of work that leverages a large real-world public dataset to pretrain models [50, 49, 54, 56].

Besides directly training image classification by DP-SGD, another direction is DP-trained generative models. The generated images can be used for classification tasks without additional privacy costs. Recent work [26] show that DP diffusion models can achieve high-quality images when pretrained on large public data like ImageNet and achieve $88.8\%$ classification accuracy for CIFAR10 at $\varepsilon = 10$.

Another line of work has shown the success of DP-SGD fine-tuning of pretrained large language models (LLMs) [28, 47, 70]. LLM pretraining can be framed as a way to learn structural priors from unstructured data [7].

**DP-SGD training from scratch.** In addition to the directly related works discussed above, the baselines for training from scratch that we compare to in this work are De et al. [15] and Sander et al. [57]. De et al. [15] make use of multiple techniques that are now a mainstay of DP-SGD training from scratch such as multiple data augmentations [32] that we also use. Sander et al. [57] propose a method for estimating the best hyperparameters for DP training at scale using smaller-scale runs. Recent works [34, 35] propose the use of inductive bias via architectural prior. They use DP-SGD to train the equivariant CNN architecture [14] and achieve $81.6\%$ at $\varepsilon = 8$ on CIFAR-10. We note that the design space of novel architectures that are especially compatible with DP is rich and mostly unexplored, and our approach is compatible with any advancements in this domain. In particular, using the architecture of Hölzl et al. [34, 35] in DP-RandP could potentially enjoy the improvements by combining the feature priors. We leave this exploration as future work.

**Two-stage training with domain-specific data.** The two-stage training of first training the classifier head and then tuning all hyperparameters has also shown to be effective for out-of-distribution (OOD) tasks [43]. Their reasoning is that if the full parameters are directly updated using the in-distribution training data, this may lead to the loss of some general features learned in the pretrained stage and result in utility drop on OOD data. Our task of DP image classification is different from the OOD task [43] but still shares some common intuition. After Phase I, our feature extractor has learned some useful priors while the classifier head has a random initialization. If we directly update the full network, too much noise is added to the full network, which may distort features learned in Phase I and lead to suboptimal performance. We also conduct experiments and find that, without noise prior (therefore model is randomly initialized), the two-stage training pipeline would not significantly improve the performance compared to directly training the full parameters (See Appendix H).

## 5 Conclusion

We leverage images generated from random processes and propose a three-phase training DP-RandP to optimize the use of noise prior. The evaluation across multiple datasets including the benchmark datasets CIFAR10 and ImageNet shows that DP-RandP can improve the performance of DP-SGD. For example, DP-RandP improves the previous best reported accuracy on CIFAR10 from $60.6\%$ to $72.3\%$ at $\varepsilon = 1$. DP-RandP is a general framework for different datasets, models, and representation learning methods. Future improvements of designs in each of these categories would potentially improve the performance of DP-RandP. DP-RandP makes use of priors from synthetic images. It would be interesting to study whether DP-RandP would improve the priors for DP with public data. Also, investigating the priors beyond image domains, e.g., language and speech tasks, for differentially private training would also be of great interest.

## Acknowledgements

We are grateful to anonymous reviewers at NeurIPS for valuable feedback. This work was supported in part by the National Science Foundation under grant CNS-2131938, the ARL's Army Artificial Intelligence Innovation Institute (A2I2), Schmidt DataX award, and Princeton E-ffiliates Award.

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

# A Ablation study for Phase I

We present our main results by using the WRN-16-4 model pretrained on Style-GAN dataset by Baradad et al. [5] with representation learning [64] for CIFAR10/CIFAR100/DermaMNIST. In this section, we provide more ablation study on the different synthetic data and different representation learning methods in Phase I. We use CIFAR10 for the evaluation.

## A.1 Results on different synthetic data

A number of synthetic datasets have been proposed by prior work. Baradad et al. [5] consider a family of synthetic datasets generated by random processes, and report that the best synthetic dataset in terms of downstream performance is generated by an untrained StyleGAN with a specific initialization, denoted as StyleGAN-Oriented. Based on this prior work, we also use StyleGAN-Oriented as our main synthetic dataset throughout the main body of the paper. We also considered the Shaders dataset proposed in Baradad et al. [6] for our ImageNet experiments. In this subsection we consider different choices of synthetic datasets and find that multiple synthetic datasets can provide good performance. That is, our results and analysis extend beyond StyleGAN-Oriented and can be applied to future proposed synthetic datasets.

We first use the feature extractor checkpoint[4] provided by Baradad et al. [5] pretrained on different synthetic datasets (Please refer Baradad et al. [5] for the full description of these datasets).

Note that Baradad et al. [5] uses a small AlexNet [42] as feature extractor, which includes Batch-Norm [36]. We freeze the feature encoder and only train the last linear model, therefore the feature extractor does not break our differential privacy guarantee. When we use the feature extractor in the main body we use WideResNet without BatchNorm.

Tab. 10 summarizes the results on different synthetic datasets. As in Baradad et al. [5], the best synthetic dataset is StyleGAN-oriented. However, even the worst-performing synthetic dataset (Dead leaves) performs similarly to the best prior work [61, 15]. Tab. 10 suggests that one potential future direction of DP-RandP is better synthetic data.

Table 10: Test accuracy (%) of private linear probing ($\varepsilon = 1$, training for 100 steps, that is the same as with WRN-16-4) on a small AlexNet trained on different synthetic images. StyleGAN-Oriented achieves the best performance. Note that all these images are generated without access to real-world images. Evaluation on CIFAR10 task.

|  | Dead leaves textures | Stat Color+WMM | Untrained StyleGAN | | | Feature Vis Dead leaves |
|---|---|---|---|---|---|---|
|  |  |  | Sparse | High freq | Oriented |  |
| Accuracy | 59.92 | 64.59 | 64.34 | 64.08 | 67.79 | 59.32 |

## A.2 Results on different representation learning methods

In Tab. 11, we compare the representation learning method of Wang and Isola [64] (results already in the main body) to MoCo [30] (we use default hyperparameters in the official repository) for use in Phase I. For Phase II and III, we use the same hyperparameters as in Tab. 13.

Similar to the main results, DP-RandP achieves significant improvements compared to baseline [15]. We find that using either of contrastive learning methods [64, 30] can achieve 72% accuracy at $\varepsilon = 1$. Also, Tab. 11 shows that DP-RandP consistently improves upon DP-RandP w/o Phase II when using either of Wang and Isola [64] or He et al. [30] in Phase I. We note that while DP-RandP is robust to the two representation learning choices of for Phase I, there is a small gap between the two methods as $\varepsilon$ increases. This suggests a future direction for further improving our method with a principled choice of contrastive learning method for Phase I.

---

[4]https://github.com/mbaradad/learning_with_noise.

Table 11: Test accuracy (%) of different representation learning methods in Phase I. Evaluation on CIFAR10 task.

| Method for Phase I | Phases | $\varepsilon = 1$ | $\varepsilon = 2$ | $\varepsilon = 3$ | $\varepsilon = 4$ | $\varepsilon = 6$ | $\varepsilon = 8$ | $\varepsilon = \infty$ |
|---|---|---|---|---|---|---|---|---|
| Wang and Isola [64] | DP-RandP | 72.32 | 77.25 | 79.99 | 81.88 | 84.01 | 85.26 | 91.69 |
| | DP-RandP w/o Phase II | 69.03 | 75.31 | 78.44 | 80.56 | 82.96 | 84.45 | 91.69 |
| MoCo [30] | DP-RandP | 72.79 | 76.26 | 78.32 | 79.78 | 81.64 | 82.84 | 90.84 |
| | DP-RandP w/o Phase II | 69.48 | 73.82 | 76.60 | 78.89 | 80.56 | 82.64 | 90.84 |
| — | Phase III only (De et al. [15]) | 56.8 | 64.9 | 69.2 | 71.9 | 71.0 | 79.5 | 88.9 |

# B   Ablation study on model architecture

We present DP-RandP on CIFAR10 with experiments with WRN-16-4 in the main body and here we also present the WRN-40-4 result on CIFAR10 in Tab. 12. The result has a similar trend as De et al. [15], WRN-40-4 can achieve better utility with more parameters. For example, at $\varepsilon = 1$, WRN-40-4 has a 0.83% increase compared to WRN-16-4. However, training WRN-40-4 model takes a longer time. Training a WRN-16-4 for 875 steps takes 5.5 hours while the same amount of steps would take 12 hours for WRN-40-4. Given the utility improvement is within 2% improvement by changing from WRN-16-4 to WRN-40-4, we use WRN-16-4 to demonstrate the effectiveness of DP-RandP for the main experiments.

Table 12: Ablation study on WRN-16-4 and WRN-40-4 on CIFAR10.

| Method | Model | $\varepsilon = 1$ | $\varepsilon = 2$ | $\varepsilon = 3$ | $\varepsilon = 4$ | $\varepsilon = 6$ | $\varepsilon = 8$ |
|---|---|---|---|---|---|---|---|
| DP-RandP w/o Phase II | WRN-16-4 | 69.03 | 75.31 | 78.44 | 80.56 | 82.90 | 84.45 |
| | WRN-40-4 | 69.45 | 76.63 | 79.64 | 82.20 | 84.51 | 85.57 |
| DP-RandP | WRN-16-4 | 72.32 | 77.25 | 79.99 | 81.88 | 84.01 | 85.26 |
| | WRN-40-4 | 73.15 | 77.53 | 80.93 | 82.83 | 85.17 | 86.12 |

# C   Experimental details

We use the Opacus library [67] for the DP-SGD implementation. Our experiments are based on the open-source code[5] of Sander et al. [57] and Baradad et al. [5]. We also provide our code. For the noise multiplier $\sigma$, given sampling rate and total step size, $\sigma$ is precomputed according to privacy loss distribution accounting as implemented in Gopi et al. [27] with epserror= 0.01 and rounded up to the precision of 0.1 to ensure that we do not underestimate the privacy loss.

**Hyperparameters.** Tab. 13, 14 and 15 summarize the hyperparameters for DP-RandP on CIFAR10, CIFAR100 and DermaMNIST respectively. We use the same total steps and batch size for CIFAR10 by following De et al. [15]. We also use the same hyperparameters of batch size and steps for CIFAR100. For DermaMNIST, we use batch size 1024 because Hölzl et al. [34] follow Klause et al. [40] and Klause et al. [40] use batch size 1024.

Table 13: Hyperparameters for DP-RandP on CIFAR10. Batch size is 4096 Augmult is 16, SGD optimizer with momentum 0 and no weight decay.

| $\varepsilon$ | 1 | 2 | 3 | 4 | 6 | 8 |
|---|---|---|---|---|---|---|
| Total Steps | 875 | 1125 | 1593 | 1687 | 1843 | 2468 |
| $\sigma$ | 9.3 | 5.6 | 4.7 | 3.8 | 2.9 | 2.6 |
| Steps in Phase II | 96 | 96 | 96 | 96 | 96 | 96 |
| LR in Phase II | 15 | 15 | 15 | 15 | 15 | 15 |
| LR in Phase III | 0.4 | 1 | 1 | 1.2 | 1.2 | 1.6 |

---

[5]https://github.com/facebookresearch/tan and https://github.com/mbaradad/learning_with_noise.

Table 14: Hyperparameters for DP-RandP on CIFAR100. Batch size is 4096 and Augmult is 16, SGD optimizer with momentum 0 and no weight decay.

| $\varepsilon$ | 3 | 4 | 6 | 8 |
|---|---|---|---|---|
| Total Steps | 1593 | 1687 | 1843 | 2468 |
| $\sigma$ | 4.7 | 3.8 | 2.9 | 2.6 |
| Steps in Phase II | 96 | 96 | 96 | 96 |
| LR in Phase II | 25 | 25 | 25 | 25 |
| LR in Phase III | 1.4 | 2 | 2.2 | 1.8 |

Table 15: Hyperparameters for DP-RandP on DermaMNIST. Batch size is 1024 and Augmult is 16, SGD optimizer with momentum 0 and no weight decay.

| $\varepsilon$ | 1 | 4 | 7.42 |
|---|---|---|---|
| Total Steps | 600 | 800 | 800 |
| $\sigma$ | 13.6 | 4.6 | 2.8 |
| Steps in Phase II | 48 | 48 | 48 |
| LR in Phase II | 2 | 2 | 2.8 |
| LR in Phase III | 0.2 | 1 | 1.2 |

## D  Additional results on DermaMNIST

We follow Hölzl et al. [34] and report the validation accuracy of DermaMNIST in Tab. 3. Here we also report the test accuracy in Tab. 16 and we can see DP-RandP outperforms the DP-SGD baseline.

Table 16: Test accuracy (%) of DP-RandP on DermaMNIST.

| Method | $\varepsilon = 1$ | $\varepsilon = 4$ | $\varepsilon = 7.42$ | $\varepsilon = \infty$ |
|---|---|---|---|---|
| DP-SGD we evaluated | $68.34_{0.28}$ | $71.08_{0.51}$ | $72.58_{0.19}$ | 76.16 |
| DP-RandP | $71.19_{0.28}$ | $73.68_{0.24}$ | $75.04_{0.25}$ | 79.70 |

## E  Privacy allocation method

We give a general privacy budget allocation strategy in Sec. 3.3. In this section, we give a detailed description of our privacy allocation strategy, the privacy ratio for CIFAR10 main results, and more results on different $\varepsilon$.

**Our privacy allocation strategy.**  Given a total number of steps $N$, we use the first $N_1$ steps to train the head classifier for Phase II, and use the remaining $N_2 = N - N_1$ steps to train the full network for Phase III. We follow Panda et al. [54] (that suggests 100 steps for linear probing) and set $N_1 = 96$ steps (this is closest to 100 steps and equals 8 epochs as each epoch contains 12 steps) in our experiment. For the x-axis in Fig. 6, we use PLD accounting as implemented in Gopi et al. [27] to calculate $\varepsilon_1$ by calculating the privacy cost of $N_1$ steps and get $\varepsilon_1/\varepsilon$ as the x-axis. Although it is known that $\varepsilon_!$ does not increase linearly with $N_1$, $\varepsilon_!$ is monotonically increasing with $N_1$ and therefore we can use this method to compute the privacy ratio of Phase II. The $N_1$ steps in Fig. 6 include [0, 12, 24, 36, 48, 96, 144, 192, 240, 288, 336, 384, 432, 480, 528, 576, 624, 672, 720, 768, 816, 864, 875] with $N = 875$.

**Privacy ratio for CIFAR10 main results.**  We visualize our privacy allocation strategy on CIFAR10 in Fig. 7, that is consistent with this strategy.

**Additional experimental results on different $\varepsilon$.**  We provide more results on different privacy allocations for different $\varepsilon$ from Tab. 17 to Tab.21. Our main results in Tab. 1 in the main paper are

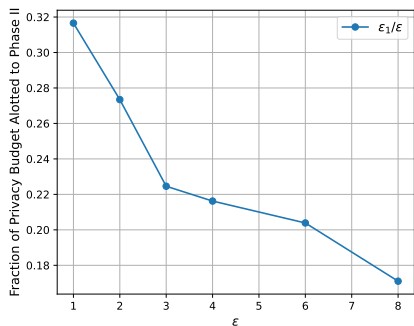

Figure 7: The fraction of total privacy budget allotted to Phase II ($\varepsilon_1/\varepsilon$) as a function of total privacy budget $\varepsilon$. As $\varepsilon$ increases, the ratio $\varepsilon_1/\varepsilon$ decreases.

produced by setting the number of epochs in Phase II to 8. From Tab. 17 to Tab.21 we can see that our privacy splitting strategy is robust to several choices of the number of epochs in Phase II (e.g., 4, 8, 12, 16) in the evaluated privacy range. Furthermore, our privacy splitting strategy is better than allocating the entire privacy budget to either Phase II (rightmost) or Phase III only (leftmost).

Table 17: Test accuracy (%) for CIFAR10 $\varepsilon = 2$. 1125 steps. 94 epochs in total.

| Epochs for Phase II | 0 | 1 | 4 | 8 | 12 | 16 | 20 | 50 | 94 |
|---|---|---|---|---|---|---|---|---|---|
| Accuracy (%) | 75.30 | 76.31 | 77.03 | 77.35 | 76.74 | 77.05 | 77.10 | 75.06 | 63.10 |

Table 18: Test accuracy (%) for CIFAR10 $\varepsilon = 3$. 1593 steps. 133 epochs in total.

| Epochs for Phase II | 0 | 1 | 4 | 8 | 12 | 16 | 20 | 50 | 100 | 133 |
|---|---|---|---|---|---|---|---|---|---|---|
| Accuracy (%) | 78.34 | 79.20 | 79.71 | 79.74 | 79.48 | 80.27 | 80.04 | 79.21 | 75.47 | 63.93 |

Table 19: Test accuracy (%) for CIFAR10 $\varepsilon = 4$. 1687 steps. 141 epochs in total.

| Epochs for Phase II | 0 | 1 | 4 | 8 | 12 | 16 | 20 | 50 | 100 | 141 |
|---|---|---|---|---|---|---|---|---|---|---|
| Accuracy (%) | 80.63 | 81.24 | 81.76 | 81.46 | 81.61 | 82.01 | 81.56 | 81.36 | 78.07 | 64.41 |

Table 20: Test accuracy (%) for CIFAR10 $\varepsilon = 6$. 1843 steps. 154 epochs in total.

| Epochs for Phase II | 0 | 1 | 4 | 8 | 12 | 16 | 20 | 50 | 100 | 150 | 154 |
|---|---|---|---|---|---|---|---|---|---|---|---|
| Accuracy (%) | 82.93 | 83.73 | 84.04 | 83.75 | 83.86 | 84.17 | 83.94 | 83.41 | 81.13 | 65.43 | 64.89 |

Table 21: Test accuracy (%) for CIFAR10 $\varepsilon = 8$. 2468 steps. 206 epochs in total.

| Epochs for Phase II | 0 | 1 | 4 | 8 | 12 | 16 | 20 | 50 | 100 | 150 | 200 | 206 |
|---|---|---|---|---|---|---|---|---|---|---|---|---|
| Accuracy (%) | 84.37 | 84.84 | 85.43 | 85.17 | 85.21 | 85.37 | 85.08 | 85.08 | 83.84 | 82.74 | 69.17 | 64.85 |

## F    Details for private linear probing on CIFAR10

For CIFAR10, we follow Baradad et al. [5] and use the alignment and uniformity loss proposed in Wang and Isola [64] to pretrain a feature extractor WRN-16-4 on StyleGAN-oriented dataset. Also we follow Baradad et al. [5] and use the third to last layer and the dimension of this layer is 4096.

Note that, the right extreme in Fig. 6 is slightly higher than $60\%$ at $\varepsilon = 1$, while the linear probing result in Tab. 4 is $67.78\%$ at $\varepsilon = 1$. This is because representation learning [64, 11] usually adds additional representation layers for representation learning and may keep it for linear probing. As mentioned earlier, we follow Baradad et al. [5] and use the third to last layer and the dimension of this layer is 4096. For Fig. 6, we keep the exact same architecture as De et al. [15] for a fair comparison where we did not use such embedding layers.

Tab. 22 summarize the hyperparameter for DP-RandP without Phase III on CIFAR10 in Sec.3.3.

Table 22: Hyperparameters for linear probing CIFAR10 experiments. Other hyperparameters include full batch size, SGD optimizer with momemtum 0.9, 100 steps, no augmentation multiplicity.

| $\varepsilon$ | 0.1 | 0.2 | 0.5 | 1 | 2 | 3 | 4 | 6 | 8 |
|---|---|---|---|---|---|---|---|---|---|
| $\sigma$ | 339 | 171 | 72 | 38 | 21 | 14 | 11 | 8 | 7 |
| LR | 0.8 | 2.2 | 5.5 | 10 | 20 | 40 | 40 | 60 | 60 |

# G   Details for ImageNet experiments

As discussed in Sec. 3.3, we achieve a new SOTA on ImageNet with additional designs for private linear probing. We first present the technical details of our method, which combines principled feature extraction and a private feature preprocessing approach. We then include the hyperparameters for our experiments on ImageNet.

## G.1   Method

Our modifications on private linear probing include principled feature extraction and a private feature preprocessing approach. We qualify that neither of these steps is novel; feature extraction variants and feature preprocessing are very standard in feature extraction pipelines.

**Pretraining.** We use a pretrained ViT-base [19] model by Yu et al. [71], that is pretrained on the Shaders-21k dataset [6] using MAE [31]. We use a different pretraining dataset here than for the simpler datasets, because models pretrained on Shaders-21k are observed to outperform those pretrained on StyleGAN [6]. Furthermore, based on our initial experiments models pretrained on StyleGAN with MoCo [30] do not perform well on ImageNet fine-tuning, only reaching $\approx 33\%$.

**Modifying the LP-FT recipe.** For the results on simpler datasets, we present a combination of linear probing and full fine-tuning (LP-FT [43]). However, we note that for more complex datasets, it is more necessary to devote privacy budget to full fine-tuning because adapting the features learned from random priors to private datasets is more challenging. Based on the increase in the best values of $\varepsilon_1$ from CIFAR10 and CIFAR100, it seems likely that full DP fine-tuning is necessary to adapt the pretrained features to ImageNet. Unfortunately, we lack the computational resources to fine-tune ViT on ImageNet with DP as prior works [15, 57] have noted that it may require hundreds or thousands of GPU-hours. Instead, we propose a *hybrid combination* of linear probing and full fine-tuning via principled feature extraction that is computationally efficient, running under 10 hours on a single A100, and also obtains better performance than the prior SOTA [57]. *We emphasize that these modifications are more for computational efficiency; if we had enough compute to do full fine-tuning of the ViT on ImageNet with sufficiently large batch size, our original DP-RandP will still work.*

**Standard feature extraction.** Standard CNNs and ResNets iteratively refine the representation of the image at each layer, and the best representation of the image is produced at the penultimate layer. The head classifier at the end of the network learns a mapping between this representation and classes. Vision transformers are slightly different; each block learns a different representation. Nonetheless, the SOTA DP fine-tuning approaches that use pretrained ViTs as feature extractors still just use the representation from the penultimate layer as input to the linear layer [54]. When we use this approach for linear probing, our best result does not exceed $33.2\%$.

**Intuition behind principled feature extraction.** The intuition behind this approach is the same as the intuition in a long line of fine-tuning approaches [48, 51, 23] and we do not claim any novelty for it. Given a sufficiently good pretrained initialization for the network [29], the domain adaptation should

have low intrinsic rank such that the adaptation from the pretrained weights to the fine-tuning weights can be modeled in a lower-dimensional subspace than that of the full model parameters [33]. Results have shown that a linearized approximation of fine-tuning can obtain competitive performance. If we add another level of approximation, if we learn a linearization of the intermediate representations of the ViT, we can model the linearization of fine-tuning.

**Principled feature extraction.** The first challenge is the dimensionality of the intermediate representations. Although it may not seem large upon initial inspection, as ViTs may only have a representation size of 768, we actually want the representation before the pooling. The representation of the input after each block in the ViT has both a temporal and feature dimension, so we pool over the temporal dimension to gather a feature map of size (4, feature size). One alternative option here is to actually learn first what weights should be used for the final linear layer, as in Evci et al. [23]. We can privatize this via the exponential mechanism. Initial results indicate this may be a very interesting direction for future work. However, this approach has a quite high computational cost, so we do not use it for the sake of reproducibility. Instead, we stride the average pooling such that the representations at each block are $4\times$ larger, and then we concatenate together the block-wise representations so that the final representation size is $4 \times$ num_blocks larger than it is in the standard feature extraction approach. For a ViT-base, num_blocks $= 12$ so this is $48\times$ larger for a final representation size and the feature size for each image is 36864.

**Feature normalization.** The first step in feature preprocessing is feature normalization. We normalize the feature vectors to a fixed norm of $C$ by the transformation below

$$x_i' = \frac{x_i \cdot C}{\|x_i\|_2}.$$

We treat $C$ as a fixed constant, and hence this normalization step doesn't result in any privacy loss about the dataset. We pick $C = 50$ because the representation multiplier from the principled feature extraction step is 48. Next we center the feature vectors around their mean $x_i = x_i - \frac{1}{|D|} \sum_{j \in D} x_j$, which requires private mean estimation. That is, the input to the training method will just be the difference between the feature and the noisy feature mean.

**Private mean estimation.** Now we introduce the motivation behind the feature normalization; so that we can do private mean estimation in high dimensions without prohibitive error rates. Given that all the feature vectors have the same dimension and fixed norm, the optimal error rate for private mean estimation will be obtained via the Gaussian mechanism. That is, we first compute the true mean and then add Gaussian noise scaled to the $\ell_2$ sensitivity of the mean. The sensitivity of the mean is $C/N$; adding or removing any datapoint can change the $\ell_2$ sensitivity by at most $C$, and there are $N$ datapoints (for ImageNet, $N = 1281167$). We do a hyperparameter search here to find the best amount of the overall privacy budget to dedicate to this step, which we can do efficiently by saving the true mean and then adding noise for each value of $\varepsilon$ we consider. We find that a very noisy estimate is sufficient, because the noise is added to the mean vector and then used to normalize all the features such that all datapoints have the same noise. This correlated noise is highly tolerable for our approach, in line with concurrent work that indicates correlated noise has much less accuracy degradation [13].

**Related work on private feature preprocessing.** One concurrent work [60] conducts a theoretical analysis that feature preprocessing provably reduces the error rate of DP linear regression and provides experiments when finetuning a model that is pretrained on ImaageNet (that is, the regime of Tab. 7. Another concurrent work [63] conducts theoretical analysis that feature preprocessing provably reduces the error rate of DP-GD from extracted features by connecting to the neural collapse regime and provides experimental validation when finetuning a model that is pretrained on ImageNet. These concurrent works provide intuition behind the success of our private feature preprocessing, although we note that neither theoretical guarantee is actually applicable to our setting because we use an entirely different method (pretraining on synthetic data, principled feature extraction, and private feature preprocessing). Our method therefore validates the theory of these concurrent works by applying private feature preprocessing in conjunction with principled feature extraction to do private linear probing on one of the most challenging datasets for DP image classification, achieving a new SOTA for methods that do not use public data.

## G.2 Results

We achieve 39.39% accuracy at $\varepsilon = 8$, and the previous SOTA [57] is 39.2% at $\varepsilon = 8$. Although this improvement is minor, we note that we have not been able to reproduce the results in Sander et al. [57] using their provided code, who themselves were not able to reproduce the result from De et al. [15] that they compare to. This reproducibility gap can be attributed to the high variance of DP training, which itself is difficult to mitigate because of the enormous computational cost required to get SOTA results on ImageNet privately. We hope that by providing our code we can bridge this gap.

We include hyperparameters of our private linear probing results on ImageNet in Tab. 23. In Tab. 23, we use $\sigma_1$ for the private mean estimation step and $\sigma_2$ for DP-SGD. Because we are using the full batch, we can use the composition theorem in Gaussian differential privacy [17] (Theorem 2.7 and Corollary 3.3) to compute the privacy loss. With $T$ steps in DP-SGD, our private linear probing is equivalent to a one-step Gaussian mechanism with noise multiplier $\sigma$, where

$$\sigma = \frac{1}{\sqrt{\frac{1}{\sigma_1^2} + \frac{T}{\sigma_2^2}}}$$

We can then compute $(\varepsilon, \delta)$-DP by computing the privacy curve of Gaussian mechanism [3, 17].

According to Panda et al. [54], as $\varepsilon$ increases, the total step size will also increase to achieve better performance. Therefore in Tab. 23, we use $T = 100$ for $\varepsilon = 1$ and $T = 200$ for $\varepsilon = 8$. In addition to Tab. 23, we also provide a few more observations during the hyperparameter search. $T = 100$ for $\varepsilon = 8$ also achieves compelling result that is close to 39%. When we fix $\varepsilon = 8$, as we increase $T$, the performance improves. Further increasing $T$ would continue to improve the performance. As the number of steps $T$ increases, the corresponding optimal learning rate would decrease. This is consistent with previous work [54].

We note that Sander et al. [57] also tried training the ViT model on ImageNet but concluded that it does not perform as well as ResNet. Our explanation for this is that ViT requires pretraining data because the architecture does not encode any natural prior, whereas CNNs naturally have a prior. The nature of convolutional filters biases CNNs to extract features with spatial locality. As we observe in Sec. 2, the impact of pretraining data is mostly at the initialization by giving the model a prior, so it stands to reason that the missing piece in utilizing ViT for DP training on ImageNet is learning a random prior from Shaders-21k [6].

Table 23: Hyperparameters for linear probing on ImageNet. $\sigma_1$ is for private mean estimation and $\sigma_2$ is for DP-SGD. Other hyperparameters include batch size = full, SGD optimizer with momentum = 0.9. For the linear layer, bias = False and zero initialization. We do not employ any additional regularization or learning rate schedule.

| $\varepsilon$ | Steps $T$ | $\sigma_1$ | $\sigma_2$ | learning rate |
|---|---|---|---|---|
| $\varepsilon = 1$ | 100 | 71 | 43 | 3 |
| $\varepsilon = 8$ | 200 | 14 | 9.33 | 10 |

## H  Two-stage training with private data

After pretraining with synthetic data, DP-RandP first training the classifier head (Phase II) and then tuning all hyperparameter (Phase III). Tab. 24 summarizes the results in the main paper and presents the comparison of our full DP-RandP, DP-RandP without Phase II and the baseline [15]. Note that DP-RandP without Phase III is not included in Tab. 24 as training the linear layer only has diminishing returns: the non-private baseline of DP-RandP w/o Phase III is 74.05%, which is worse than DP-RandP w/o Phase II at $\varepsilon = 2$. Tab. 24 shows the importance of combining both Phase II and Phase III in DP-RandP.

Table 24: The importance of phases in DP-RandP. Evaluation of test accuracy (%) on CIFAR10.

| Phases | $\varepsilon = 1$ | $\varepsilon = 2$ | $\varepsilon = 3$ | $\varepsilon = 4$ | $\varepsilon = 6$ | $\varepsilon = 8$ |
|---|---|---|---|---|---|---|
| DP-RandP | 72.32 | 77.25 | 79.99 | 81.88 | 84.01 | 85.26 |
| DP-RandP w/o Phase II | 69.03 | 75.31 | 78.44 | 80.56 | 82.96 | 84.45 |
| Phase III only (De et al. [15]) | 56.8 | 64.9 | 69.2 | 71.9 | 71.0 | 79.5 |

Our DP-RandP uses synthetic data for pretraining to give a warm initialization for two-stage training with private data. Tab. 25 shows the two-stage training with private data with random initialization. As noted in Tab. 13, we use $\sigma = 9.3$ for $\varepsilon = 1$ while De et al. [15] use $\varepsilon = 10$ for the same number of 875 steps (equals to 73 epochs). This is because we only round $\sigma$ up to 0.1 and we ensure the $\sigma$ we use in Tab. 13 will not exceed the designed privacy bound. As we add less noise compared to De et al. [15], the baseline result, i.e., directly updating the full parameters during training (0 epoch for Phase II), is 57.46.

Table 25: Test accuracy (%) of two-stage training with private data by random initialization on CIFAR10.

| Epochs for Phase II | 0 | 1 | 2 | 4 | 8 | 12 | 16 | 20 | 50 | 73 |
|---|---|---|---|---|---|---|---|---|---|---|
| Accuracy (%) | 57.46 | 57.01 | 56.27 | 55.04 | 54.45 | 54.09 | 53.15 | 52.26 | 47.00 | 25.80 |
| Std. | 0.48 | 0.69 | 0.57 | 0.90 | 0.80 | 0.41 | 0.91 | 0.97 | 1.53 | 2.08 |

Tab. 25 shows that when the feature extractor is randomly initialized, if we train the head classifier with more steps while keeping the same total steps, the performance will decrease compared to directly finetuning the whole network. Our analysis of this is that, the feature extractor is not encoded with any prior and it is better to update the full network in the whole training procedure to learn more information.

