# OpenReview forum: "Differentially Private Image Classification by Learning Priors from Random Processes"
_NeurIPS.cc/2023/Conference — NeurIPS 2023 spotlight_

### Official Review · Reviewer_dEgE · 2023-06-18

**Soundness:** 2 fair
**Presentation:** 4 excellent
**Contribution:** 3 good
**Rating:** 7
**Confidence:** 4

**Summary:**

This paper presents a method for synthesizing pre-training data for private fine-tuning without requiring access to any real data. The data generation process utilizes some fundamental properties of natural images that have been previously studied in fields outside of private learning. For example, natural images often exhibit a high degree of sparsity and contain oriented structures. After pre-training the model on the synthesized data, the authors employ a two-stage private learning framework. The first stage involves tuning only the linear classifier, while the second stage tunes all parameters.

**Strengths:**

1. The concept of generating 'free' pre-training data is intriguing, and the methods employed in this paper appear reasonable.

2. Experiments conducted on several vision datasets demonstrate that such synthesized data is important in private learning, particularly when the privacy budget is small.

3. The authors introduce a two-stage approach for private learning, which is new to the private learning community and results in some improvements.

4. The results are well-organized and easy to follow.


**Weaknesses:**

1. As shown in Figure 6, some improvements of the proposed algorithm are from the two-stage approach (Phase-II and Phase-III) for private learning. Can the two-stage approach improve the baseline? E.g., can the method in Hölzl et al., [27] get better results if it also uses Phase-II and Phase-III for private learning and set the hyperparameters accordingly?

2. Kumar et al., 2022 [35] propose to first tune the classification head and then tune all parameters. The authors may want to discuss how their motivation can also explain the benefit of the two-stage approach, as a complementary to the discussion in Line 102 – Line 114.

I am willing to increase my score if the authors can address my concerns.

**Questions:**

1. Is it easy to generate ‘free’ pre-training data for language or speech tasks? Language or speech data also exhibits some natural structures, e.g., the subject–verb–object word order. However, I’m not sure whether there are off-the-shelf algorithms that use such properties to generate data.

2. I see the authors mark 'Reproducibility' as yes but the code is not uploaded. Is there a plan to make the code that reproduces all the main observations publicly available?

**Limitations:**

Limitations are discussed adequately.

---

> ### Author Rebuttal · Authors · 2023-08-09
>
> We would like to thank the reviewer for the insightful feedback. We appreciate the reviewer recognizing our empirical improvements, as well as the importance of synthetic data and two-phase training.
>
> 1. “Can the two-stage approach improve the baseline? E.g., can the method in Hölzl et al., [27]”
>
> We incorporate DP-RandP with the baseline by De et al.[13] which uses the WideResNet architecture with GroupNorm layer and show that our DP-RandP can achieve significant improvement compared to De et al.[13]. Our DP-RandP is compatible with advancements in synthetic data, architecture, and representation learning. As discussed in Section 4, “we could use the architecture of Hölzl et al.[26,27] in our method and potentially enjoy the improvements of their method stacked on top of our own improvements by combining the feature priors”. As currently the source code of Hölzl et al.[20] is not available, we defer incorporation of our DP-RandP with their architecture design as future work. We look forward to investigating whether applying the equivariant neural network and our framework for learning from priors can further improve the utility once the authors release their code.
>
> 2. “Discussion on Kumar et al.”
>
> As discussed in Kumar et al.[21], the two-stage approach of first fine-tuning the classifier head and then tuning all hyperparameters has shown to be effective in previous studies[22,23]. Kumar et al.[21] show that such an approach is also effective for out-of-distribution (OOD) tasks. Their reasoning is that if the full parameters are updated using the in-distribution training data in the fine-tuning stage, then the full network would be more likely to overfit to the in-distribution data. This may lead to the loss of some general features learned in the pretrained stage and result in inferior performance on OOD data. They therefore utilize this two stage framework to solve this problem for OOD data.
>
> Our task of DP image classification is different from the OOD task in Kumar et al.[21] but still shares some common intuition. The training data and test data in our experiments follow the same distribution. While there is a large distribution shift between synthetic data generated by StyleGAN-oriented and real data in CIFAR10, training on the CIFAR10 training data should not cause utility drop in CIFAR10 test data. We conduct experiments and find that the two-stage pipeline would not significantly improve the performance compared to directly training the full parameters if no noise is added during training. After stage one, our feature extractor has learned some useful information while the classifier head has a random initialization. If we directly update the full network, too much noise is added to the full network, which may distort features learned in Phase I and lead to suboptimal performance. Panda et al [16] show that the linear head classifier is robust to noise and can be trained in less than 100 steps. We therefore first apply finetuning to the classifier head with only a few steps (96 steps for CIFAR10, which is a small privacy cost compared to the total privacy budget) and then tune all hyperparameters. We will add this discussion in the revised paper.
>
> 3. ”pre-training data for other domain”
>
> Please see CR#2 for a discussion on pre-training data for other data modalities.
>
> 4. ”Reproducibility”.
>
> Please see CR#3 for the anonymous code link.

---

> > ### Comment · Reviewer_dEgE · 2023-08-14
> >
> >
> > Thank you for your response. It addresses most of my concerns. There is one remaining question, and I am happy to increase my score if the authors can address it.
> >
> > 1. Can the two-stage approach improve the baseline?
> >
> > There seems to be a misunderstanding of my question. What I mean is the following: 1) skip Phase-I, i.e., do not pre-train the model. 2) use Phase-II and Phase-III training anyway. For example, the authors can take the code of De et al. [13], and first train the head classifier on top of random layers. After training the head classifier, the authors can continue the Phase-III training. I would be happy to increase my score if the authors are willing to conduct this experiment and tune how the privacy budget is allocated (as in Figure 6).

---

> > > ### Author Response · Authors · 2023-08-17
> > > **Two-stage approach for baseline**
> > >
> > > We thank reviewer for the insightful feedback. Here we provide our Phase II+Phase III results on CIFAR10 at $\varepsilon=1$. Each reported result is averaged across five independent runs and we also report the stand derivation.
> > >
> > > As noted in Table 11, we use $\sigma=9.3$ for $\varepsilon=1$ while De et al. use $\sigma=10$ for the same number of 875 steps (equals to 73 epochs). This is because we only round $\sigma$ up to 0.1 and we ensure the $\sigma$ we use in Table 11 will not exceed the designed privacy bound. As we add less noise compared to De et al., the baseline result is $57.46_{0.48}$.
> > >
> > >
> > > Epochs for Phase II |$0$ |$1$ | $2$| $4$ | $8$ | $12$ | $16$ | $20$ | $50$ | $73$
> > > --- |---|---|----|-----|-----|------|------|------|------|-----
> > > Accuracy |$57.46_{0.48}$|$57.01_{0.69}$|$56.27_{0.57}$|$55.04_{0.90}$|$54.45_{0.80}$|$54.09_{0.41}$|$53.15_{0.91}$|$52.26_{0.97}$|$47.00_{1.53}$|$25.80_{2.08}$
> > >
> > > This result shows that when the feature extractor is randomly generated, if we train the head classifier with more steps while keeping the same total steps, the performance will decrease compared to directly finetuning the whole network. Our analysis of this is that, the feature extractor is not encoded with any prior and it is better to update the full network in the whole training procedure to learn more information. We will add this result and the discussion in the draft.

---

> > > > ### Comment · Reviewer_dEgE · 2023-08-18
> > > >
> > > > Thank you for the experiment. It addresses my concern. I have changed my recommendation to Accept.

---

### Official Review · Reviewer_mcxP · 2023-07-04

**Soundness:** 4 excellent
**Presentation:** 3 good
**Contribution:** 3 good
**Rating:** 6
**Confidence:** 4

**Summary:**

The paper focuses on the problem of improving the privacy-utility trade-offs for the foundational technique of DP-SGD. The authors explore gains from learning priors from images generated by random processes and transferring the priors to private data. They do this via designing DP-RandP, a three-phase approach that first learns priors, then training a head classifier on top of frozen features, and then training both the feature extractor and classifier. Through an empirical evaluation on CIFAR10, CIFAR100, and MedMNIST, they show that their method is able to achieve new SOTA accuracies when training from scratch.

**Strengths:**

1. The paper focuses on the important problem of improving the privacy-utility trade-offs for the foundational technique of DP-SGD.
2. The paper is well-written, and the novel ideas are easy to understand.


**Weaknesses:**

Minor point:
1. I think it might be cleaner to mention the gains in privacy cost in terms of the absolute gains in the parameter used to measure privacy. E.g., in Fig 2 caption, since $\epsilon$ goes from 6 to 3, it might be easier to understand a factor of 2 gain? Stating a factor of 20 (which technically applies to the change in the outcome probabilities used in the defn. of DP) might be somewhat misleading (and overstated). Same comment for the gains stated in line 188.


**Questions:**

Since the novel technique has 2 new phases compared to prior works (e.g., De et al. [13]), it might be useful to a reader to understand what gains the first stage provides vs. gains provided by the second phase by itself? Although this is somewhat studied in section 3.3, I feel it might be useful to understand in the evaluation in section 3.2 (so DP-RandP can be compared with existing baselines, and with specific phases provided to those baselines).


**Limitations:**

Yes.

---

> ### Author Rebuttal · Authors · 2023-08-09
>
> We thank the reviewer for the valuable feedback. We appreciate the reviewer recognizing our empirical improvements on the important problem.
>
> 1. “privacy gain”.
>
> Thanks for the suggestion. We will change the statement to reducing privacy cost from epsilon=6 to epsilon=3 in the revised paper (similar change to line 188).
>
> 2. ”providing results for understanding the gains of three phases in section 3.2”.
>
> We will add Tab.17 for CIFAR10 in the revised paper. The result on CIFAR10 is already in the paper and here we summarize them in a single table for better comparison. This table compares our full DP-RandP and DP-RandP without Phase II. We did not include DP-RandP w/o Phase III in this table as the linear layer has diminishing returns: the non-private baseline of DP-RandP w/o Phase III is 74.05%, which is worse than DP-RandP w/o Phase II at epsilon=2.
>
> In addition, we have also tried DP-RandP w/o Phase I, which does not include the representation learning phase with synthetic data. The results of such a setting are worse than that in De et al.[13]. The explanation is that, when the model is randomly initialized, there is no additional utility gain to first fix the feature extractor and train the linear layer, as the feature extractor does not contain any useful information and it is better to update the full network in the whole training procedure.

---

> > ### Comment · Reviewer_mcxP · 2023-08-15
> > **Thank you for the rebuttal**
> >
> > I have read the authors' rebuttal, and thank them for answering the questions in my review. It would be nice to see the details provided in the response to the 2nd point above included somewhere in the paper (even if in the Appendix) so practitioners can be more aware of the interplay of the different phases of the proposed technique as well as the synergies between them. I would like to maintain my positive recommendation for the paper.

---

> > > ### Author Response · Authors · 2023-08-18
> > >
> > > Thank you for your comment. To followup on your question on what gains are provided by the second phase by itself, here we provide our Phase II+Phase III results on CIFAR10 at $\varepsilon=1$. Each reported result is averaged across five independent runs and we also report the standard derivation. By excluding Phase I in this experiment, we can answer the question of whether there are any gains provided by the second phase by itself.
> > >
> > > As noted in Table 11, we use $\sigma=9.3$ for $\varepsilon=1$ while De et al. use $\sigma=10$ for the same number of 875 steps (equivalent to 73 epochs). This is because we only round $\sigma$ up to 0.1 and we ensure the $\sigma$ we use in Table 11 will not exceed the designed privacy bound. As we add less noise compared to De et al., the baseline result is $57.46_{0.48}$.
> > >
> > >
> > > Epochs for Phase II |$0$ |$1$ | $2$| $4$ | $8$ | $12$ | $16$ | $20$ | $50$ | $73$
> > > --- |---|---|----|-----|-----|------|------|------|------|-----
> > > Accuracy |$57.46_{0.48}$|$57.01_{0.69}$|$56.27_{0.57}$|$55.04_{0.90}$|$54.45_{0.80}$|$54.09_{0.41}$|$53.15_{0.91}$|$52.26_{0.97}$|$47.00_{1.53}$|$25.80_{2.08}$
> > >
> > > This result shows that when the feature extractor is randomly generated, if we train the head classifier with more steps while keeping the same total steps, the performance will decrease compared to directly finetuning the whole network. Our analysis of this is that, the feature extractor is not encoded with any prior and it is more optimal to allocate privacy budget towards updating the full network in the whole training procedure to learn more information. We will add this result and the discussion in the draft.
> > >
> > > As a final point, because the discussion period is drawing to a close, we are interested in hearing any further concerns you may have.

---

### Official Review · Reviewer_BUrr · 2023-07-05

**Soundness:** 3 good
**Presentation:** 3 good
**Contribution:** 3 good
**Rating:** 6
**Confidence:** 4

**Summary:**

The authors propose using synthetic data to reduce the utility degradation induced by DP-SGD when training on image classification datasets. They suggest a 3 steps procedure: 1) pre-training on synthetic data, 2) DP training the linear head, 3) DP fine-tuning the full body. The procedure seems effective on the considered benchmarks and at various levels of $\epsilon$.

**Strengths:**

- Although the novelty is limited (the synthetic data was already designed by other works and the idea of pre-training to facilitate convergence in DP-SGD training has been explored), the observation this strategy is particularly effective in DP training is interesting and will be surely extremely useful for future research in DP training for image classification.
- The paper is clearly written, the experiments are well explained.
- The analyses carried out are interesting, and provide practitioners insights on how to adapt the proposed  training procedure to their setup


**Weaknesses:**

- The considered datasets are relatively simple. The method does not seem to scale well as the number of classes increases (see CIFAR-100 and ImageNet experiments). While this holds true for most DP training procedures, the ImageNet experiment might suggest the methodology has little or marginal utility in this case.
- While the comparison with ScatterNet and Equivariant-CNNs is particularly lengthy in the Related Works section, a comparison with methodologies leveraging public non-synthetic data (e.g. [1,2,3,4]) is missing. While such a comparison is unfair, and will show inferior performance of the proposed methodology, this does not diminish its utility. The purpose of such a comparison would be to show the gap that exists between using synthetically generated data (both data-driven or not) and real data, and would set the goal of bridging this gap with purely synthetic techniques in future works.

Minor:
- The formatting of the captioning of when multiple subfigures are present on the same line does not render very well, I'd recommend improving it.

[1] How to make semi-private learning more effective, https://openreview.net/pdf?id=vVXRNcltT6

[2] Differentially Private Image Classification from Features, https://arxiv.org/pdf/2211.13403.pdf

[3] Large Scale Transfer Learning for Differentially Private Image Classification, https://arxiv.org/pdf/2205.02973.pdf

[4] Differentially private diffusion models generate useful synthetic images, https://arxiv.org/pdf/2302.13861.pdf


**Questions:**

- Did the authors consider the idea of augmenting their training set with fractals (e.g. [5], and not only black-and-white ones, but also coloured ones)
- Did the authors consider applying other contrastive or self-supervised pre-training strategies to better leverage the synthetic data? It may be possible that different strategies can differently impact the accuracy of the linear probe (e.g. as shown in [4]) and may attain higher utility in Phase 3 of the training procedure.
- Could you add error bars to the plots? (e.g. $+/- 2\sigma$)

[5] FractalDB, https://github.com/hirokatsukataoka16/FractalDB-Pretrained-ResNet-PyTorch

**Limitations:**

Properly Addressed.

---

> ### Author Rebuttal · Authors · 2023-08-09
>
> We thank the reviewer for the constructive feedback. We’re encouraged that the reviewer appreciates our empirical improvements, analysis and insights.
>
> 1. ”...seem to not scale well…”
>
> We first provide new and improved results on ImageNet; we achieve 39.35% accuracy at epsilon=8, and previous SOTA[12] is 39.2% at epsilon=8. First, as in our previous results, we consider a model pretrained on [25]; in this case the model is a ViT-base. If we do Phase II with extracted features, we achieve the same 33.2% at epsilon=8. We now explain the steps we took to improve our performance. We make two key modifications over the two-phase approach we use for other image tasks. We emphasize that these modifications are more for computational efficiency; if we had enough compute to do full fine-tuning of the ViT on ImageNet with sufficiently large batch size, our original two-phase approach will still work.
>
> Our first modification is to approximate full fine-tuning with linear probing on larger feature representations that we create by aggregating intermediate representations from the model. The representation of the input after each block in the ViT has both a temporal and feature dimension, so we pool over the temporal dimension to gather a feature map of size (4, feature size). The feature size is 768 for ViT-base with 12 blocks, so the representation size is 4\*12\*768, 48 times larger than the original representation we would have otherwise extracted. Our motivation is that because there is a large domain shift between [25] and ImageNet, the final representation may not be sufficient to learn the task. Fine-tuning is necessary to potentially lift up intermediate representations from earlier in the network. By doing linear probing on these much larger features, we can also take advantage of intermediate representations.
>
> The second modification is to approximate the work of the normalization layer that we would update during full fine-tuning of the entire ViT, by manually normalizing the features. To do this we first normalize each feature vector to a fixed norm of 50, as the feature vector is 48 times larger than the original representation. There may be better normalization schemes, but we did not have enough time to tune this choice. We next privately estimate the mean over the entire ImageNet feature vector dataset, using the Gaussian mechanism with a privacy cost of epsilon=0.1, and subtract the private mean from all feature vectors. This is equivalent to doing non-private centering and then adding the same Gaussian noise to the entire dataset. This procedure can be thought of as a one-time approximation to the normalization layer that is known to speed up training by centering data.
>
> The hyperparameters we use for linear probing are: lr = 10, batch size = full, epochs = 200, optimizer = SGD, momentum = 0.9, bias = False, initialization = zero. We do not employ any additional regularization or learning rate schedule.
>
> We note that [12] also tried training the ViT model on ImageNet but concluded that it does not perform as well as ResNet. Our explanation for this is that ViT requires pretraining data because the architecture does not encode any natural prior, whereas CNNs naturally have a prior. As we observe in the main paper, the impact of pretraining data is mostly at the initialization by giving the model a prior, so it stands to reason that the missing piece in utilizing ViT for DP training on ImageNet is pretraining on [25].
>
> We agree with the reviewer that there is still a large gap between the private result and non-private result on tasks with more classes like CIFAR-100 and ImageNet. This is due to the inherent difficulty of differentially-private training on these complex tasks (even outside the context of our innovations). Our work aims to close this gap by leveraging synthetic data to learn priors and utilizing our three phase framework. DP-RandP still achieves >10% improvements on CIFAR100 for the evaluated epsilons compared to the DP-SGD baseline. Future work in this direction to further improve the performance could include better synthetic data, model architecture, representation learning techniques for Phase I to learn better priors, better algorithms beyond our DP-RandP.
>
> 2. “comparison with public data”
>
> Please see CR#1.
>
> 3. “format of caption”.
>
> We will fix this in the revised paper.
>
> 4. “augmentation with fractals”
>
> Baradad et al.[6] studied FractalDB[17] as a baseline and showed that StyleGAN-oriented is better than FractalDB for a range of tasks, though Baradad et al.[6] did not explicitly state the fractals is colored or black-and-white. Kataoka et al.[17] investigated the impact of colorization and achieved 93.1% accuracy (w/ color) vs. 92.9% accuracy (w/o color). Based on [16,17], possible improvements by current colored fractals for DP image classification tasks might be marginal. Future work on designing better fractal methods could be helpful in improving such tasks.
>
> 5. “other contrastive learning“
>
> In Tab.18 we compare the contrastive learning method of [18] (results already in the submission) to MoCo[19] (we use default hyperparameters in the official repo) for use in Phase I. For Phase II and III, we use same hyperparameters as in Tab.11. We will include Tab.18 in the revised paper.
>
> Similar to the main results, DP-RandP achieves significant improvements compared to baseline[13]. We find that using either of [18,19] can achieve 72% accuracy at epsilon=1. Also, Tab.18 shows that DP-RandP consistently improves upon DP-RandP w/o Phase II when using either of [18,19] in Phase I.
> We note that while DP-RandP is robust to the choice of [18,19] for Phase I, there is a small gap between the choice of [18,19] as epsilon increases. This suggests a future direction for further improving our method with a principled choice of contrastive learning method for Phase I.
>
> 6. “...Error bars”.
>
> We currently report std. in the tables. We will add error bars in the plots.

---

> > ### Comment · Reviewer_BUrr · 2023-08-10
> > **Thank you for your rebuttal**
> >
> > I have read the rebuttal, and I find it addresses most of my concerns.
> >
> > - On performance: Although the change to get the improvement for ImageNet is significant, I think it is acceptable as it does not change the key concept of the paper.  Obviously, the used $\epsilon$ values are quite high, and tight $\epsilon$ would be preferred. However, this is acceptable as I understand it would be difficult to show improvements in extremely tight privacy regimes. Please include all the details in the updated draft and the source code.
> >
> > - On the comparison with semi-private methods: including a discussion in the updated draft could be useful.
> >
> > - About fractals and other contrastive learning techniques: Thanks for the clarification. Since these are natural questions that may arise in a reader, it could be useful to include some of the responses in the updated draft.

---

> > > ### Author Response · Authors · 2023-08-11
> > > **Thank you**
> > >
> > > We are grateful our rebuttal answered most of the concerns in the review. As recommended by the reviewer, we will include these points in the updated draft, as we cannot update the OpenReview draft during the review period.

---

> > > ### Author Response · Authors · 2023-08-18
> > >
> > > We thank you again for your review and will include all these clarifications in the camera ready. As the discussion period is drawing to a close, we would like to ask if you have any remaining concerns that we may address. If you feel that our rebuttal has answered your concerns, we would be grateful if you might consider increasing your score.

---

### Official Review · Reviewer_t5fK · 2023-07-25

**Soundness:** 3 good
**Presentation:** 3 good
**Contribution:** 4 excellent
**Rating:** 8
**Confidence:** 4

**Summary:**

The paper considers the problem of training ML models in a privacy-preserving way, focusing primarily on image classification. The main observation in this paper is that synthetic data learned from random processes can be used to learn useful visual priors which can be extremely useful for private training. Building on this, the authors propose a three phase approach for privately training ML models: 1. Learn a prior (feature extractor) from synthetic images generated by random processes, 2. Train the head classifier with frozen features with privacy parameter epsilon_1, and 3. End-to-end training using the remaining privacy budget epsilon_2. The authors provide an extensive empirical evaluation of this method over three different datasets for a wide range of privacy parameters, demonstrating that their methods significantly improve over state-of-the-art results for all privacy regimes. For example, for CIFAR10 with epsilon = 1, they obtain 72% accuracy, significantly improving over the SOTA (60%).


**Strengths:**


As written above, the paper has some clear and strong empirical improvements in the performance of private training for image classification. Moreover, the idea of using priors learned from synthetic data is interesting and turns out to be extremely useful for private training as demonstrated in the paper. This techniques could also be useful for private training in other domains.

**Weaknesses:**

In general, the paper has strong results, and aside from some minor comments (see below), my main critique is that the paper explains the algorithm too quickly without too much details, and starts the empirical evaluations too early. I would recommend the authors to spend more time in explaining their algorithms, and giving more background for phase 1 which is basically the main component of the algorithm. Also, the authors should give enough details for future readers to be able to replicate the results; for example, which version of DP-SGD is used? What is the batch size? Does it use momentum?

More comments:

1. While the methods in the paper improve over SOTA for private training, they don’t achieve the same performance as private training with public data. I would suggest that the authors also compare their methods with methods that use a public dataset for pretaining (the authors currently do this only for epsilon = 1).

2. The idea of having both phase 2 and 3 is interesting and it would be useful to see the improvements of that when combined with pretraining using public data.

3. It would be useful to provide further evidence for the importance of combining both the second and third approached in the algorithm: one possibility is to compare the 3-phased approach with an approach that uses phase 1+2 or phase 1+3. This has been done in separate tables in the paper but providing the performance of all three approaches in the same plot would help better illustrate the advantages of all three phases.

4. Privacy splitting:
    - The privacy splitting strategy used in the paper is not completely clear and should be explained in more detail.

    - The paper only shows how privacy splitting affects the performance for small values of eps=1 in Figure 6. It would be useful to provide a similar figure for larger values of epsilon, to show the advantages of phase 3 over phase 2 for such epsilons, and to show how robust is this method to privacy splitting.

5. Can this approach be applied beyond image classification?

**Questions:**

See above.

**Limitations:**

See above.

---

> ### Author Rebuttal · Authors · 2023-08-09
>
> We first want to thank the reviewer for appreciating both the novelty and empirical success of our method, and furthermore the potential application to other domains.
>
> 1. “Presentation and experiment details.”
>
> For the experimental set-up, we use the DP-SGD introduced by Abadi et al.[11]. We use the opacus library for DP-SGD and our code is built based on the codebase by Sander et al.[12]. We mostly follow the hyperparameters suggested by De et al. [13] (batch size = 4096, momentum=0). We include our hyperparameters in Appendix C. We have open sourced the code in a comment to the ACs as per the NeurIPS rebuttal guidelines. We will improve the explanation of the algorithm and background in Phase I in the revised version as suggested.
>
> 2. “Comparison to private training with public data”.
>
> Please see CR#1 where we provide a quantitative comparison to methods that use public data.
>
> 3. “Phase II and Phase III for public data setting.”
>
> Given limited rebuttal time (more time is needed on the task with public data for a fair comparison, which is not studied in this submission), we did not have the chance to conduct experiments in Phase II and Phase III for public data. Here we briefly discuss our intuition for why Phase II and Phase III should also help for the public data setting. In the non-private scenario, self-supervised learning has already shown to be effective for large amounts of data [14] (especially when no label information is available). Table 5 in Yu et al. [15] shows that using public data with random labels to pretrain the model can achieve better utility for DP-SGD compared to the ground truth label, which suggests that the feature information in the public data is more important. Our Phase I is trained with feature information and does not need label information. Recent works [2,3,16] show that pretraining the model on the large public data and finetuning a linear layer with private data can achieve good performance on a wide range of tasks, which suggest that the linear layer is robust to the noise. For the pretrained model, Panda et al. [16] use the backbone model pretrained by Dosovitskiy et al.[14] (Mehta et al. [2,3] do not explicitly mention the pretraining method). We follow the intuition that the linear layer is robust to the noise, and we train our head classifier in Phase II and full network in Phase III. Considering the findings in [2,3,14,15,16], our framework should also help in the public data setting when there is a gap between linear probing and full finetuning. If the linear probing method is already pretty good as in [2,3], we also suggest directly applying linear probing.
>
> 4. “It would be useful to provide further evidence for the importance of combining both the second and third approached in the algorithm”
>
> We thank the reviewer for the suggestion. We will add Tab.17 in the revised paper. This table compares our full DP-RandP and DP-RandP without Phase II. We did not include DP-RandP w/o Phase III in this table as training the linear layer only has diminishing returns: the non-private baseline of DP-RandP w/o Phase III is 74.05%, which is worse than DP-RandP w/o Phase II at epsilon=2.
>
> 5. “Privacy splitting method “
>
> a. “Explanation for privacy splitting method”
>
> Given a total number of steps N, we use the first N1 steps to train the head classifier for Phase II, and use the remaining N2 = N-N1 steps to train the full network for Phase III. We follow Panda et al.[16] (that suggests 100 steps for linear probing) and set N1 = 96 steps (this is closest to 100 steps and equals to 8 epochs as each epoch contains 12 steps) in our experiment. For the x-axis in Figure 6, we use PLD accounting as implemented in [24] to calculate $\epsilon_1$ by calculating the privacy cost of N1 steps and get $\epsilon_1$/$\epsilon$ as the x-axis. Although it is known that $\epsilon_1$ does not increase linearly with N1, $\epsilon_1$ is monotonically increasing with N1 and therefore we can use this method to compute the privacy ratio of Phase II. We will include this explanation in the revised paper.
>
> b.”Experiments for different epsilons.”
>
> Here we provide more results for $\varepsilon \in [2, 3, 4, 6, 8]$ in Tab.19-23. Due to the limited time and computation constraints, we are not able to calculate error bars for these results. Our main results in Tab.1 in the main paper are produced by setting the number of epochs in Phase II to 8. From Tabs.19-23 we can see that our privacy splitting strategy is robust to several choices of the number of epochs in Phase II (e.g., {4, 8, 12, 16}) in the evaluated privacy range. Furthermore, our privacy splitting strategy is better than allocating the entire privacy budget to either Phase II or Phase III only. We will add these new results in our revised paper.
>
> 6. “Other domains beyond image classification.”
>
> Please see CR#2 for a discussion on extending our method to other data modalities.

---

> > ### Comment · Reviewer_t5fK · 2023-08-14
> > **Response**
> >
> > I'm happy with the authors' response and still recommend to accept the paper.

---

### Author Rebuttal · Authors · 2023-08-09

We thank the reviewers for their constructive comments and suggestions. We’re encouraged that reviewers appreciate our empirical improvements, novelty, analysis, insights and potential impacts. We provide responses to common comments below. Tab.1-15 are in the current submission and Tab.16- 23 are included in the uploaded PDF. Due to space limitation, references are included as comment after common response.

1. “Comparison to private training with public data”.

We have a paragraph discussion on “DP with public data” in Section 4, where we briefly compare our method with Nasr et al. [1] at epsilon=1. Here we include the comparison of our method to previous works [1,2,3,4,5] by considering different public data settings and different epsilons in Tab.16 (quantitative comparison with [1,2,3]). We will add Tab.16 and discussion in our revised paper.

Our DP-RandP is comparable with Nasr et al.[1], which in fact utilizes a limited amount of in-distribution data as public data for pre-training. This comparison supports the promise of our DP-RandP as mentioned in the paper “the prior learned from images generated from random processes can help as much as the prior learned from limited in-distribution public data.”

There is also another line of work that leverages a large real-world public dataset to pretrain models [2,3,4]. Mehta et al. [2,3] can achieve > 95% accuracy even for epsilon=1 by pretraining on ImageNet. Pinto et al.[4] studied projecting the data into top-k principal components at epsilon<=1 and can achieve 81.3% accuracy at epsilon=0.1 using ResNet-50 pretrained on ImageNet as the feature extractor. Beside directly training image classification by DP-SGD, another direction is DP-trained generative models. The generated images can then be used for classification tasks without additional privacy costs. Recent work [5] showed that DP diffusion models can achieve high-quality images when pretrained on large public data like ImageNet. For example, [5] can achieve 88.8% classification accuracy for CIFAR10 at epsilon=10. We acknowledge that there is still a gap between our DP-RandP and these works (since they have access to real public data). Closing the gap between leveraging synthetic data and leveraging large-scale real public data is an interesting direction of future work (some possible ideas could be: better synthetic data, model architecture, representation learning algorithm for Phase I to learn better priors, and better algorithms beyond our DP-RandP).

2. ”Other domain beyond image classification”

We thank reviewers for this very interesting question! DP-RandP significantly enhances the state-of-the-art by utilizing the synthetic data generated from StyleGAN-oriented in image classification tasks. We believe that synthetic data can be generated with human knowledge in other domains as well to enhance differentially-private training. We provide a preliminary discussion on speech tasks and language modeling tasks below.

a. Speech tasks.

Speech data shares some similar properties to image data. The image data has a space domain and a frequency domain. The speech data has a time domain and a frequency domain. The synthetic data by Baradad et al.[6] is designed based on known characteristics of the frequency domain of images. It is also possible to generate the speech data in the frequency domain based on frequency knowledge of syllables for natural speech and then transform it back to the time domain. We could also incorporate the subject–verb–object word order in the time domain for synthesis as suggested by the reviewer dEgE. As for the training method, recent works by Chiu et al. [7] also showed that the self-supervised learning helps speech recognition tasks. These findings shed light on future applicability of our method in speech recognition tasks.

b. Language tasks.

As suggested by the reviewer dEgE, subject–verb–object word order could be designed for text synthesis. We could also encode the fact that synonyms should be close to each other in embeddings. Self-supervised learning methods such as BERT[8] and Roberta[9] are already widely studied in NLP tasks. We could generate several n-grams which do not have privacy risks for the language modeling tasks. Furthermore, Shi et al.[10] showed that first finetuning the model with private text removed corpus can help the followed differentially-private training. Although this work [10] utilizes more than synthetic data, we believe it provides insight into the feasibility and future applicability of our method in language tasks.

3. “Reproducibility”.

We will open source our code. We include the link to anonymous codebase in a separate comment to Area Chair by following NeurIPS rebuttal guideline.

---

> ### Author Response · Authors · 2023-08-10
> **Please use this reference list for the referred paper numbers in the rebuttals.**
>
> References:
>
> [1] M. Nasr, S. Mahloujifar, X. Tang, P. Mittal, and A. Houmansadr. Effectively using public data in privacy preserving machine learning. ICML 2023.
>
> [2] H. Mehta, A. G. Thakurta, A. Kurakin, and A. Cutkosky. Towards large scale transfer learning for differentially private image classification. TMLR 2023.
>
> [3] H. Mehta, W. Krichene, A. G. Thakurta, A. Kurakin, and A. Cutkosky. Differentially private image classification from features. TMLR 2023.
>
> [4] F. Pinto, Y. Hu, F. Yang, A. Sanyal, How to make semi-private learning more effective, TrustML-(un)Limited, ICLR 2023.
>
> [5] S. Ghalebikesabi, L. Berrada, S. Gowal, I. Ktena, R. Stanforth, J. Hayes, S.De, S. Smith, O. Wiles and B. Balle. Differentially private diffusion models generate useful synthetic images, arXiv preprint arXiv:2302.13861.
>
> [6] M. Baradad, J. Wulff, T. Wang, P. Isola, and A. Torralba. Learning to see by looking at noise. NeurIPS 2021.
>
> [7] C.Chiu, J. Qin, Y. Zhang, J. Yu, Yonghui Wu. Self-Supervised Learning with Random-Projection Quantizer for Speech Recognition. ICML 2022.
>
> [8] J. Devlin, M. Chang, K. Lee, K. Toutanova, BERT: Pre-training of Deep Bidirectional Transformers for Language Understanding. NAACL 2019.
>
> [9] Y. Liu, M. Ot, N. Goyal, J. Du, M. Joshi, D. Chen, O. Levy, M. Lewis, L. Zettlemoyer, V. Stoyanov. RoBERTa: A Robustly Optimized BERT Pretraining Approach. arXiv preprint arXiv:1907.11692.
>
> [10] W. Shi , R. Shea , S. Chen, C. Zhang , R. Jia, Z. Yu. Just Fine-tune Twice: Selective Differential Privacy for Large Language Models. EMNLP 2022.
>
> [11] M. Abadi, A. Chu, I. Goodfellow, H. B. McMahan, I. Mironov, K. Talwar, and L. Zhang. Deep learning with differential privacy. In Proceedings of the 2016 ACM SIGSAC Conference on Computer and Communications Security, pages 308–318. ACM, 2016.
>
> [12] T. Sander, P. Stock, and A. Sablayrolles. Tan without a burn: Scaling laws of dp-sgd, ICML 2023.
>
> [13] S. De, L. Berrada, J. Hayes, S. L. Smith, and B. Balle. Unlocking high-accuracy differentially private image classification through scale. arXiv preprint arXiv:2204.13650.
>
> [14] A. Dosovitskiy, L. Beyer, A. Kolesnikov, D. Weissenborn, X. Zhai, T. Unterthiner, M. Dehghani, M. Minderer, G. Heigold, S. Gelly, J. Uszkoreit, and N. Houlsby. An image is worth 16x16 words: Transformers for image recognition at scale, ICLR 2021.
>
> [15] D. Yu, H. Zhang, W. Chen, and T.-Y. Liu. Do not let privacy overbill utility: Gradient embedding perturbation for private learning. ICLR 2021.
>
> [16] A. Panda, X. Tang, V. Sehwag, S. Mahloujifar, and P. Mittal. Dp-raft: A differentially private recipe for accelerated fine-tuning. arXiv preprint arXiv:2212.04486.
>
> [17] H. Kataoka, K. Okayasu, A. Matsumoto, E. Yamagata, R. Yamada, N. Inoue, A. Nakamura, and Y. Satoh. Pre-training without natural images, IJCV 2021.
>
> [18] T. Wang and P. Isola. Understanding contrastive representation learning through alignment and uniformity on the hypersphere. ICML 2020.
>
> [19] K. He, H. Fan, Y. Wu, S. Xie, R. Girshick. Momentum Contrast for Unsupervised Visual Representation Learning, CVPR 2020.
>
> [20] F. A. Hölzl, D. Rueckert, and G. Kaissis. Equivariant differentially private deep learning, arXiv preprint arXiv:2301.13104.
>
> [21] A. Kumar, A. Raghunathan, R. M. Jones, T. Ma, and P. Liang. Fine-tuning can distort pretrained features and underperform out-of-distribution, ICLR 2022.
>
> [22] S. Levine, Chelsea Finn, Trevor Darrell, and P. Abbeel. End-to-end training of deep visuomotor policies, JMLR 2016.
>
> [23] F. Kanavati and M. Tsuneki. Partial transfusion: on the expressive influence of trainable batch norm parameters for transfer learning. In Medical Imaging with Deep Learning, 2021
>
> [24] S. Gopi, Y. T. Lee, and L. Wutschitz. Numerical composition of differential privacy. NeurIPS 2021.
>
> [25] M. Baradad, R. Chen, J. Wulff, T. Wang, R. Feris, A. Torralba, and P. Isola. Procedural image programs for representation learning. NeurIPS 2022.

---

### Decision · Program_Chairs · 2023-09-21

**Decision:**

Accept (spotlight)

**Comment:**

This paper proposes a novel differentially private training method that substantially improves the state-of-the-art for DP training on widely researched datasets such as CIFAR-10.

There is an agreement among the reviewers to accept this paper and recommend it for a spotlight.